# EquiJump: Protein Dynamics Simulation via SO(3)-Equivariant Stochastic Interpolants

## Abstract

Mapping the conformational dynamics of proteins is crucial for elucidating their functional mechanisms. While Molecular Dynamics (MD) simulation enables detailed time evolution of protein motion, its computational toll hinders its use in practice. To address this challenge, multiple deep learning models for reproducing and accelerating MD have been proposed drawing on transport-based generative methods. However, existing work focuses on generation through transport of samples from prior distributions, that can often be distant from the data manifold. The recently proposed framework of stochastic interpolants, instead, enables transport between arbitrary distribution endpoints. Building upon this work, we introduce EquiJump, a transferable SO(3)-equivariant model that bridges all-atom protein dynamics simulation time steps directly. Our approach unifies diverse sampling methods and is benchmarked against existing models on trajectory data of fast folding proteins. EquiJump achieves state-of-the-art results on dynamics simulation with a transferable model on all of the fast folding proteins.

## 1 Introduction

Proteins are the workhorses of the cell, and simulating their dynamics is critical to biological discovery and drug design (Karplus and Kuriyan, 2005). Molecular Dynamics (MD) simulation is an important tool that leverages physics for time evolution, enabling precise exploration of the conformational space of proteins (Hollingsworth and Dror, 2018). However, sampling with physically-accurate molecular potentials requires small integration time steps, often making the simulation of phenomena at relevant biological timescales prohibitive (Lane et al., 2013).

To tackle this challenge, several studies have adopted deep learning models to capture surrogates of MD potentials and dynamics (Noé et al., 2020; Durumeric et al., 2023; Arts et al., 2023). More recent works (Schreiner et al., 2023; Li et al., 2024; Jing et al., 2024) have proposed to use deep learning-based simulators trained on long-interval snapshots of MD trajectories to predict future states given some starting configuration. These models draw from neural transport models (Ho et al., 2020; Lipman et al., 2022), learning a conditional or guided bridge between a prior distribution ($\rho_0 = \mathcal{N}$) and the target data manifold of simulation steps ($\rho_1 = \rho_{\text{data}}$). In contrast to this, the recent paradigm of Stochastic Interpolants (Albergo et al., 2023a; Albergo and Vanden-Eijnden, 2023) provides a method for directly bridging distinct arbitrary distributions.

In this work, we utilize this framework and introduce **EquiJump**, a Two-Sided Stochastic Interpolant model which bridges between long-interval timesteps of protein simulation directly (Figure 1). EquiJump is SO(3)-equivariant and simulates all heavy atoms directly in 3D. We train a transferable model on 12 fast-folding proteins (Majewski et al., 2023; Lindorff-Larsen et al., 2011) and successfully recover their dynamics.

Our main contributions are as follows:

- We extend the Two-Sided Stochastic Interpolants framework to simulate the dynamics of three-dimensional representations. By training on trajectory data, our approach directly leverages the close relationship between consecutive timesteps.

- We benchmark our model against existing generative frameworks for simulating the long-time dynamics of a large protein, demonstrating the increased accuracy and parameter efficiency of our approach.

- We train *transferable* simulators for capturing the dynamics of 12 fast-folding proteins, comparing it to existing force-field model, ablating the impact of model complexity on simulation quality, and evaluating the trade-off between sampling speed and accuracy.

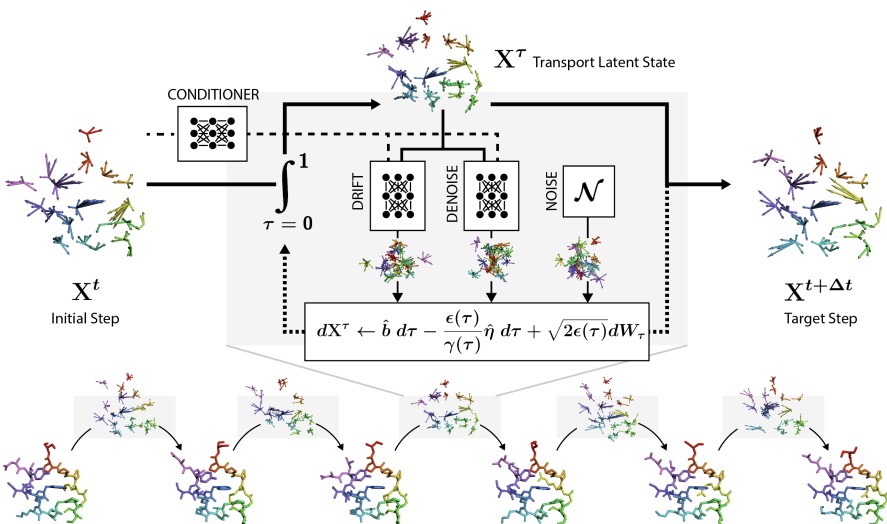

Figure 1: **Direct bridging of 3D all-atom simulation time steps**: EquiJump runs an stochastic interpolants-based transport process on coordinates and 3D geometric representations to generate future time frames from an initial state. Gray boxes depict transport across the learned latent space, which takes Gaussian noise perturbations and uses noise ($\hat{\boldsymbol{\eta}}$) and drift ($\hat{\boldsymbol{b}}$) predictions to directly transform all-atom proteins across time and 3D space.

## 2 RELATED WORK

Recent advancements in protein modeling through deep learning have led to the development of several models capable of replicating molecular dynamics (MD) trajectories. (Wang et al., 2019; Husic et al., 2020) introduced supervised models trained through direct force matching, demonstrating their ability to transfer simulations across different proteins. The most common machine learning approach to atomistic MD are interatomic potentials trained on forces (Satorras et al., 2021; Batatia et al., 2022). Still, force matching methods require force information and in practice are often limited to short time steps. Methods like (Du et al., 2024) can help sampling the transition path between two states, but rely on existing potentials. Several recent approaches reconstruct generic trajectories without forces. Many of them are based on an equivariant backbone: (Xu et al., 2023) and (Wu et al., 2023) both directly predict the next configuration based on several previous steps, and have been applied to general tasks as well as MD simulations. The latter was tested on a small protein dataset and shown to reconstruct deterministic motion. Other equivariant models use diffusion to match distributions: (Han et al., 2024) aims at modeling the whole trajectory and can reconstruct MD, while (Luo et al., 2024) specializes on tasks such as structural relaxation. (Zhang et al., 2024) is based on flow matching and predicts the next configuration from previous frames, but was not applied to MD tasks. EquiJump is similar to these approaches, instead drawing on Two-Sided Stochastic Interpolants and specialized for protein dynamics. In particular, we leverage the statistical nature of our approach to reproduce the stochastic nature of the task.

Drawing on coarse-graining and force matching, (Majewski et al., 2023) implements a unified transferable model for multiple proteins. (Fu et al., 2023) learns to predict accelerated, coarse grained dynamics of polymers with GNNs. (Köhler et al., 2023) utilizes Normalizing Flows (Gabrié et al., 2022) for coarse grained force-matching, while (Arts et al., 2023) builds upon Denoising Diffusion

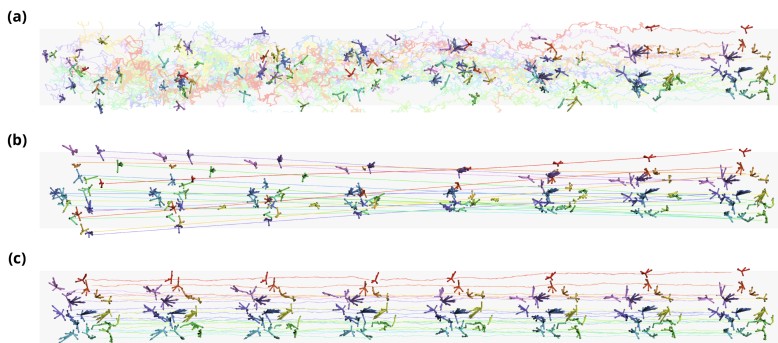

Figure 2: **Neural Transport of Tensor Clouds**. **(a) DDPM** defines an SDE for denoising samples from a Gaussian prior, while standard **(b) Flow Matching** traces a velocity field-based ODE for moving the Gaussian samples. **(c) Two-Sided Stochastic Interpolants** instead enable transporting through a local, normally-perturbed latent space that remains close to the manifold of the data.

Probabilistic Models (DDPM) (Ho et al., 2020) through Graph Transformers (Shi et al., 2020; Costa, 2021). These approaches focus heavily on coarse-grained representations, which limit their ability to simulate the full complexity of protein dynamics at the all-atom level. In contrast, EquiJump operates directly at the all-atom scale, achieving efficiency through SO(3)-equivariant neural networks for processing residue representation.

Recent models have proposed generating samples from a prior distribution while conditioning on an initial configuration. Timewarp (Klein et al., 2023) enhances MCMC sampling with conditional normalizing flows, while ITO (Schreiner et al., 2023) uses a PaiNN-based network (Schütt et al., 2021) to learn a conditional diffusion model for next-step prediction. Similarly, F$^3$low (Li et al., 2024) employs FramePred (Yim et al., 2023) and Optimal Transport Guided Flow Matching (Zheng et al., 2023). Finally, (Jing et al., 2024) applies one-sided stochastic interpolants (Ma et al., 2024) to interpolate or extrapolate molecular configurations. These approaches rely on transforming Gaussian priors via stochastic (SDE) or ordinary differential equations (ODE), where the prior often lies far from the true data distribution. Instead, EquiJump uses two-sided stochastic interpolants to directly bridge trajectory snapshots, leveraging the configuration proximity of consecutive timesteps and enabling a transport that stays close to physical states (Figure 2; Appendix A.1).

## 3 METHODS

### 3.1 STOCHASTIC INTERPOLANTS

Neural transport methods have demonstrated outstanding performance in generative tasks (Ma et al., 2024; Liu et al., 2023; Lipman et al., 2022; Ho et al., 2020). Stochastic Interpolants (Albergo et al., 2023a; Albergo and Vanden-Eijnden, 2023) are a recently proposed class of generative models that have reached state-of-the-art results in image generation (Ma et al., 2024; Albergo et al., 2023b). One-sided stochastic interpolants, which generalize flow matching and denoising diffusion models, transport samples from a prior distribution $\mathbf{X}_0 \sim \mathcal{N}$ to a target data distribution $\mathbf{X}_1 \sim \rho_1$ by utilizing latent variables $\mathbf{Z} \sim \mathcal{N}$ through the stochastic process $\{\mathbf{X}_\tau\}$:

$$\mathbf{X}_\tau = J(\tau, \mathbf{X}_1) + \alpha(\tau)\mathbf{Z} \qquad (1)$$

where $\tau \in [0, 1]$ is the time parameterization. The interpolant function $J$ satisfies boundary conditions $J(0, \mathbf{X}_1) = 0$ and $J(1, \mathbf{X}_1) = \mathbf{X}_1$, and the noise schedule $\alpha$ satisfies $\alpha(0) = 1$ and $\alpha(1) = 0$.

In contrast, two-sided stochastic interpolants enable learning the transport from $\mathbf{X}_0 \sim \rho_0$ to $\mathbf{X}_1 \sim \rho_1$ when $\rho_0$ and $\rho_1$ are arbitrary probability distributions (Figure 2). Two-sided interpolants are described by the stochastic process $\{\mathbf{X}_\tau\}$:

$$\mathbf{X}_\tau = I(\tau, \mathbf{X}_0, \mathbf{X}_1) + \gamma(\tau)\mathbf{Z} \qquad (2)$$

where $\tau \in [0, 1]$ and to ensure boundary conditions, the interpolant $I$ and noise schedule $\gamma$ must satisfy the following: $I(0, \mathbf{X}_0, \mathbf{X}_1) = \mathbf{X}_0$ and $I(1, \mathbf{X}_0, \mathbf{X}_1) = \mathbf{X}_1$, and $\gamma(0) = \gamma(1) = 0$.

The probability $p(\tau, \mathbf{X})$ of a stochastic interpolant satisfies the transport equation:

$$\partial_\tau p(\tau, \mathbf{X}) + \nabla \cdot (b(\tau, \mathbf{X})p(\tau, \mathbf{X})) = 0 \tag{3}$$

and the boundary conditions $p(0, \mathbf{X}) = p_0$ and $p(1, \mathbf{X}) = p_1$. Here, $b(\tau, \mathbf{X})$ is the expected velocity:

$$b(\tau, \mathbf{X}) = \mathbb{E}\left[\partial_\tau \mathbf{X}_\tau \mid \mathbf{X}_\tau = \mathbf{X}\right] = \mathbb{E}\left[\partial_\tau I(\tau, \mathbf{X}_0, \mathbf{X}_1) + \partial_\tau \gamma(\tau)\mathbf{Z} \mid \mathbf{X}_\tau = \mathbf{X}\right] \tag{4}$$

We can similarly define the noise term $\eta(\tau, \mathbf{X})$ as:

$$\eta(\tau, \mathbf{X}) = [\mathbf{Z} \mid \mathbf{X}_\tau = \mathbf{X}] \tag{5}$$

In practice, the exact forms of $b$ and $\eta$ are not known for arbitrary distributions $p_0$, $p_1$, and are thus parameterized by neural networks. (Albergo et al., 2023a) shows that we can learn the functions $\hat{b} \approx b$ and $\hat{\eta} \approx \eta$ by optimizing:

$$\min_{\hat{b}} \int_0^1 \mathbb{E}\left[\frac{1}{2}\hat{b}(\tau, \mathbf{X}_\tau)^2 - (\partial_\tau I(\tau, \mathbf{X}_0, \mathbf{X}_1) + \partial_\tau \gamma(\tau)\mathbf{Z}) \cdot \hat{b}(\tau, \mathbf{X}_\tau)\right] d\tau \tag{6}$$

$$\min_{\hat{\eta}} \int_0^1 \mathbb{E}\left[\frac{1}{2}\hat{\eta}(\tau, \mathbf{X}_\tau)^2 + \mathbf{Z} \cdot \hat{\eta}(\tau, \mathbf{X}_\tau)\right] d\tau \tag{7}$$

We can then sample $\mathbf{X}_{\tau=1} \sim p(\tau = 1, \mathbf{X}_1)$ through an ordinary differential equation (ODE), or a stochastic differential equation (SDE):

$$d\mathbf{X}_\tau = \hat{b}(\tau, \mathbf{X}_\tau)d\tau \tag{8}$$

$$d\mathbf{X}_\tau = \left(\hat{b}(\tau, \mathbf{X}_\tau) - \frac{\epsilon(\tau)}{\gamma(\tau)}\hat{\eta}(\tau, \mathbf{X}_\tau)\right) d\tau + \sqrt{2\epsilon(\tau)}dW_\tau \tag{9}$$

where $W_\tau$ is the Weiner process. Once we have learned the expected velocity $\hat{b}$ and noise $\hat{\eta}$, the above equations can be integrated numerically starting from $(\tau = 0, \mathbf{X}_0 \sim p_0)$ to $(\tau = 1, \mathbf{X}_1 \sim p_1)$. Furthermore, following from eqs. (8) and (9) the probability $p(\tau, \mathbf{X}_\tau)$ is SO(3)-equivariant when $\hat{b}$ and $\hat{\eta}$ are SO(3)-equivariant and $dW_\tau$ is isotropic. We provide more details on interpolant parameterization in Appendix A.2.

## 3.2 TWO-SIDED STOCHASTIC INTERPOLANTS FOR DYNAMICS SIMULATION

We extend the two-sided stochastic interpolant framework to learn a time evolution operator from trajectory data $[\mathbf{X}^t]_{t=1}^L$. Given a source time step $\mathbf{X}^t$ and its consecutive target step $\mathbf{X}^{t+1}$, we define the distribution boundaries of our interpolant as $\rho_0 = \rho(\mathbf{X}^t)$ and $\rho_1 = \rho(\mathbf{X}^{t+1} \mid \mathbf{X}^t)$. The conditional nature of the target distribution requires that our predictions for drift $\hat{b}$ and noise $\hat{\eta}$ are explicitly conditioned on the source step $\mathbf{X}^t$. We apply this approach to simulating all-atom protein dynamics, as depicted in Figure 1. In this context, $\mathbf{X}^t$ represents a 3D all-atom protein conformation at time $t$, which is provided as input to our model. We frame it as the source distribution, and set $\mathbf{X}_{\tau=0} = \mathbf{X}^t$. We then employ an iterative process governed by the integration of eqs. (8) and (9) from $\tau = 0$ to $\tau = 1$. This produces a sample $\mathbf{X}_{\tau=1}$, which follows the distribution $\mathbf{X}_{\tau=1} \sim \rho_1$, generating a next step in the simulation $\mathbf{X}^{t+1}$.

## 3.3 MULTIMODAL INTERPOLANTS OF GEOMETRIC REPRESENTATIONS

We treat data $\mathbf{X}$ represented as geometric features positioned in three-dimensional space, $\mathbf{X} = [(\mathbf{V}_i, \mathbf{P}_i)]_{i=1}^N$, which we refer to as the *Tensor Cloud* representation (Figure 2). In this formulation, each $\mathbf{V}_i$ is a tensor of irreducible representations (irreps) of O(3) or SO(3), associated with a 3D coordinate $\mathbf{P}_i \in \mathbb{R}^3$. The feature representations $\mathbf{V}$ are arrays of irreps up to order $l_{max}$ where for each $l \in [0, l_{max}]$, the tensor $\mathbf{V}^l$ represents geometric features with dimensions $\mathbf{V}^l \in \mathbb{R}^{H \times (2l+1)}$, where $H$ denotes the feature multiplicity.

We extend interpolant eqs. (8) and (9) to the multi-modal type $\mathbf{X}_i = (\mathbf{V}_i, \mathbf{P}_i)$ by integrating geometric features and coordinate components as $d\mathbf{X}_i^\tau = (d\mathbf{V}_i^\tau, d\mathbf{P}_i^\tau)$. For computing the losses eqs. (6) and (7), we define the Tensor Cloud dot product as $\mathbf{X}_i \cdot \mathbf{X}_j = \mathbf{V}_i \cdot \mathbf{V}_j + \mathbf{P}_i \cdot \mathbf{P}_j$. In general, treating the feature and coordinate components independently allows for different parameterizations of the interpolant. In this work, we use the same interpolant form for both components, only adjusting the variances $(\sigma_\mathbf{V}^2, \sigma_\mathbf{P}^2)$ in sampling the variable $\mathbf{Z}$. For further details see Appendix A.2.

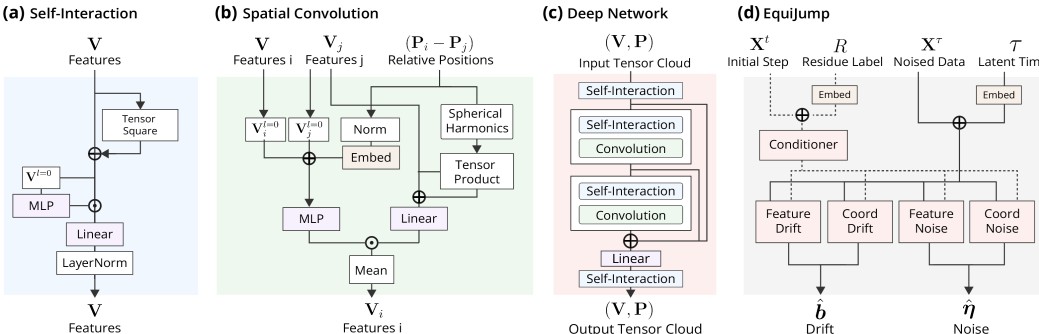

Figure 3: **EquiJump Architecture**: **(a)** The **Self-Interaction** Layer updates geometric features independently, mixing $\mathbf{V}^l$ of different degrees into new features through a Tensor Square operation. **(b)** The **Spatial Convolution** layer updates representations by aggregating the tensor product of neighbors messages with the spherical harmonics embedding of the relative 3D vector between the positions of those neighbors. **(c)** We stack the above modules to form a block, and build a base network out of $L$ blocks for making predictions. **(d)** A shared conditioner and 4 headers are built from the base network. The conditioner processes sequence and the current simulation step, producing latent embeddings that are fed to the prediction headers. The headers independently predict features and coordinates updates for drift and noise components of the stochastic process.

### 3.3.1 PROTEIN STRUCTURE REPRESENTATION

We represent a protein monomer $(\mathbf{R}, \mathbf{X})$ as a sequence $\mathbf{R}$ and a Tensor Cloud $\mathbf{X}$. Our model is designed to update $\mathbf{X}$ while being conditioned on $\mathbf{R}$. Each residue $i$ consists of three components: a residue label $\mathbf{R}_i \in \mathcal{R} = \{\text{ALA, GLY}, \dots \}$, the $C_\alpha$ 3D coordinate $\mathbf{P}_i^\alpha \in \mathbb{R}^3$, and a geometric feature of order $l = 1$ with multiplicity 13, $\mathbf{V}_i^A \in \mathbb{R}^{13 \times 3}$. This feature encodes the relative 3D vector from the $C_\alpha$ to all other heavy atoms in the residue, following a canonical ordering. For residues with fewer than 13 non-$C_\alpha$ heavy atoms, we pad the atom vectors. This modeling approach, based on (King and Koes, 2020; Costa et al., 2024), allows for the direct representation of all heavy atoms in 3D, while maintaining a coarse-grained representation anchored on the $C_\alpha$.

### 3.3.2 NEURAL NETWORK ARCHITECTURE AND TRAINING

To efficiently process 3D data, we utilize established modules of Euclidean-equivariant neural networks (Geiger and Smidt, 2022; Miller et al., 2020). We design our network to predict the drift $\hat{b} = (\hat{b}_{\mathbf{V}}, \hat{b}_{\mathbf{P}})$ and noise $\hat{\eta} = (\hat{\eta}_{\mathbf{V}}, \hat{\eta}_{\mathbf{P}})$ terms, conditioned on the sequence $\mathbf{R}$, the source structure $\mathbf{X}^t$, the latent transport structure $\mathbf{X}_\tau^t$, and the latent time $\tau$ (Figure 3). The EquiJump layer is built from two SO(3)-equivariant modules: the Self-Interaction (Figure 3.a) module for updating features $\mathbf{V}^l$ independently from coordinates, based on (Costa et al., 2024; Batatia et al., 2022); and the Spatial Convolution (Figure 3.b) module for sharing information between neighbors, based on Tensor Field Networks (Thomas et al., 2018). At each layer, we also employ residual connections and the SO(3)-equivariant layer norm from (Liao and Smidt, 2022). We build a deep neural network (DNN) by stacking $L$ times the above blocks (Figure 3.c). Refer to Appendix A.2 for additional details.

We use 5 DNNs in our model (Figure 3.d): 1 conditioner network $\mathbf{f}_{\text{cond}}$ and 4 header networks for predicting each of $\hat{b}_{\mathbf{V}}, \hat{b}_{\mathbf{P}}, \hat{\eta}_{\mathbf{V}}, \hat{\eta}_{\mathbf{P}}$ independently. Given a configuration $\mathbf{X}^t$, we first prepare a hidden representation $\tilde{\mathbf{X}}^t = \mathbf{f}_{\text{cond}}(\mathbf{R}, \mathbf{X}^t)$. The 4 headers take $(\tilde{\mathbf{X}}^t, \mathbf{X}_\tau^t, \tau)$ to produce predictions of each component of the drift $\hat{b}$ and the noise $\hat{\eta}$. For efficiency, the embedding $\tilde{\mathbf{X}}^t$ is made independent of $\tau$, and only the prediction headers are used in the integration loop

of the latent transport. We train and sample these networks following Algorithms 1 and 2:

---

**Algorithm 1** EquiJump Training

**Require:** Sequence $\mathbf{R}$
**Require:** Trajectory Data $[\mathbf{X}^t]_{t=1}^T$
**Require:** Interpolant Parameters $I_\tau, \gamma(\tau)$
**Require:** Networks $\hat{b}_\mathbf{V}, \hat{b}_\mathbf{P}, \hat{\eta}_\mathbf{V}, \hat{\eta}_\mathbf{P}, \mathbf{f}_{\text{cond}}$
1: $t \sim \mathcal{U}(1, T-1)$
2: $\tau \sim \mathcal{U}(0, 1)$
3: $\mathbf{Z}^\tau \sim \mathcal{N}(0, \mathbb{I})$
4: $\tilde{\mathbf{X}}^t \leftarrow \mathbf{f}_{\text{cond}}(\mathbf{R}, \mathbf{X}^t)$
5: $\mathbf{X}_\tau^t \leftarrow (1-\tau) \cdot \mathbf{X}^t + \tau \cdot \mathbf{X}^{t+1} + \gamma(\tau)\mathbf{Z}^\tau$
6: $\hat{\eta} \leftarrow (\hat{\eta}_\mathbf{V}(\tilde{\mathbf{X}}^t, \mathbf{X}_\tau^t, \tau), \hat{\eta}_\mathbf{P}(\tilde{\mathbf{X}}^t, \mathbf{X}_\tau^t, \tau))$
7: $\hat{b} \leftarrow (\hat{b}_\mathbf{V}(\tilde{\mathbf{X}}^t, \mathbf{X}_\tau^t, \tau), \hat{b}_\mathbf{P}(\tilde{\mathbf{X}}^t, \mathbf{X}_\tau^t, \tau))$
8: **Gradient Step**
9: $-\nabla \left( \frac{1}{2}\|\hat{b}\| - \hat{b} \cdot \left( \partial_\tau I_\tau(\mathbf{X}^t, \mathbf{X}^{t+1}) + \dot{\gamma}(\tau) \cdot \mathbf{Z}^\tau \right) \right.$
$\left. + \frac{1}{2}\|\hat{\eta}\| - \hat{\eta} \cdot \mathbf{Z}^\tau \right)$

---

**Algorithm 2** EquiJump Sampling

**Require:** Sequence $\mathbf{R}$
**Require:** Start Step $\mathbf{X}^t$
**Require:** Interpolant Parameters $\epsilon(\tau), \gamma(\tau)$
**Require:** Networks $\hat{b}_\mathbf{V}, \hat{b}_\mathbf{P}, \hat{\eta}_\mathbf{V}, \hat{\eta}_\mathbf{P}, \mathbf{f}_{\text{cond}}$
**Require:** Integration Timestep $d\tau$
1: $\mathbf{X}_{\tau=0}^t \leftarrow \mathbf{X}^t$
2: $\tilde{\mathbf{X}}^t \leftarrow \mathbf{f}_{\text{cond}}(\mathbf{R}, \mathbf{X}^t)$
3: **for** $(\tau \leftarrow 0 ; \tau < 1 ; \tau \leftarrow \tau + d\tau)$ **do**
4: $\quad \mathbf{Z}^\tau \sim \mathcal{N}(0, \mathbb{I})$
5: $\quad \hat{\eta} \leftarrow (\hat{\eta}_\mathbf{V}(\tilde{\mathbf{X}}^t, \mathbf{X}_\tau^t, \tau), \hat{\eta}_\mathbf{P}(\tilde{\mathbf{X}}^t, \mathbf{X}_\tau^t, \tau))$
6: $\quad \hat{b} \leftarrow (\hat{b}_\mathbf{V}(\tilde{\mathbf{X}}^t, \mathbf{X}_\tau^t, \tau), \hat{b}_\mathbf{P}(\tilde{\mathbf{X}}^t, \mathbf{X}_\tau^t, \tau))$
7: $\quad d\mathbf{X}^\tau \leftarrow (\hat{b} - \frac{\epsilon(\tau)}{\gamma(\tau)}\hat{\eta})d\tau + \sqrt{2\epsilon(\tau)}\mathbf{Z}^\tau$
8: $\quad \mathbf{X}_{\tau+d\tau}^t \leftarrow \mathbf{X}_\tau^t + d\mathbf{X}_\tau^t$
9: **return** $\mathbf{X}_{\tau=1}^t$

---

# 4 EXPERIMENTS AND RESULTS

## 4.1 FAST-FOLDING PROTEINS

To evaluate the capability of our model in reproducing protein dynamics, we leverage the dataset of 12 fast-folding proteins produced by (Majewski et al., 2023), and originally investigated in (Lindorff-Larsen et al., 2011). The dataset consists of millions of snapshots of MD for 12 proteins ranging from 10 to 80 residues, where in each trajectory snapshots are taken at the rate of 100 ps. The trajectories are made up of several NVT runs (20 to 100 ns) at $T = 350$ K from different starting configurations sampling the phase space, for an aggregated simulation time of hundreds to thousands of $\mu$s per protein. We refer to the original work and Appendix A.3 for more details.

While sampling uniformly on the whole dataset is effective in producing configurations within the original phase space, it is not efficient for learning the potential energy surface. For the slowest modes of the system, states are very high in free energy and heavily under-represented in our training dataset. To address this issue, taking inspiration by classical sampling methods such as umbrella sampling (Torrie and Valleau, 1977) and metadynamics (Laio and Parrinello, 2002) we propose a reweighing of the training set. Notably, since our model learns $\rho(\mathbf{X}^{t+1} \mid \mathbf{X}^t)$, the target distribution remains unchanged as long as we fix the endpoints $(\mathbf{X}^t, \mathbf{X}^{t+1})$: this sampling only varies the speed at which different parts of the phase space are learned. To reweight our sampling towards dynamically informative states, we first find relevant degrees of freedom through TICA analysis (Pérez-Hernández et al., 2013), which offers a reduced dimensional space that highlights the slow macroscopic modes of the system. We then fit a small number of clusters (Figure 8) through k-means in this simplified space. Our enhanced dataset first samples a cluster, then from the cluster a configuration and its transition. Refer to Appendix A.4 for more details.

## 4.2 EQUILIBRATION OF MODEL DYNAMICS

To study the long-term dynamical behavior of our models, we estimate the stationary distribution of the learned dynamics and apply a correction to the density of sample observables. We leverage Time-lagged Independent Component Analysis (TICA) (Pérez-Hernández et al., 2013) and build Markov State Models (MSM) on clusters over TIC components. We obtain reference TICA components from the original trajectories by considering a similarity based on the Euclidean distance between $C_\alpha$ and a lagtime of 2ns. To estimate long-term probabilities after equilibration, we reweight the density of sampled configurations. We first cluster the configuration using K-means in the first 4 TIC dimensions with 100 clusters. We then build a Markov State Model (MSM) on the basis of these clusters by estimating the transition matrix at long time-lag (45 to 95ns). Finally, from the MSM largest eigenvectors we obtain the steady state probability of each cluster, which we use to

reweight the distribution of observables for approximating their behavior at dynamical equilibrium. Note that this reweighing is extremely sensitive to the correct description of the transition states, as a wrong sampling of the transition probability will exponentially affect the relative probability of different basins. As such, the correct description of these probability distributions is a good indicator of the faithfulness of the long-term dynamics.

### 4.3 GENERATIVE TRANSPORT COMPARISON

We conduct a comparative analysis of our method against baseline methods for protein simulation (Figure 2), evaluating their ability to capture the long-term dynamics of Protein G. We compare against DDPM (Ho et al., 2020) as applied in (Arts et al., 2023; Schreiner et al., 2023), Flow Matching (Lipman et al., 2022) as used in (Li et al., 2024), and One-Sided Interpolants (Albergo et al., 2023a). For this evaluation, we vary the latent variable noise variance of positions $\sigma_{\mathbf{P}}^2 = \{1, 3, 5\}$ while keeping the features noise variance fixed at $\sigma_{\mathbf{V}}^2 = 1$. For comparison, we adapt our network and only use 2 headers (noise or drift) for DDPM and Flow Matching. We train all models with $H = 32$ for 200k steps and batch size 128. For each model, we sample 1000 trajectories of 500 steps (50 ns) with 100 steps of latent variable integration. In Figure 4, we show the free energies (MSM-reweighted, as in 4.2) along the first two TIC components for the best performing models. In Table 1, we compare the Jensen-Shannon divergence (Lin, 1991) of observables against reference for each model.

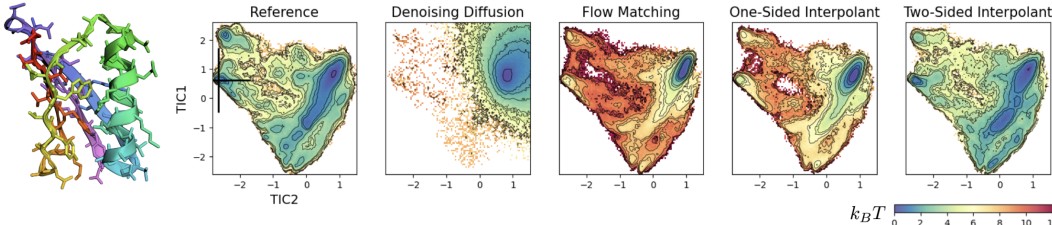

Figure 4: **Protein G and Free Energies on its TIC components for different models of Generative Simulation**. (Left) Protein G crystal. (Right) Estimated free energies on the first TIC components for samples produced by DDPM, Flow Matching and Stochastic Interpolants. We observe Two-Sided Interpolants outperform other transport in recovering the TICA profile.

This analysis demonstrates that the local transport of Two-Sided Interpolants is well-suited for capturing the consecutive step dynamics of MD trajectories. While all models can locate a primary basin of the conformational landscape of the protein, standard DDPM and Flow Matching struggle to capture the slower and wider, less probable dynamics. We find that our approach is particularly adept at capturing these subtleties. Specifically, direct transport between timesteps (Two-Sided) outperforms transport methods from a prior distribution (One-Sided). Our results indicate that Two-Sided Interpolants are a robust tool for learning accurate forward-time simulation operators for Molecular Dynamics.

| $\sigma_{\mathbf{P}}^2$ | DDPM | | | Flow Matching | | | One-Sided Interpolant | | | EquiJump | | |
|---|---|---|---|---|---|---|---|---|---|---|---|---|
| | 1 | 3 | 5 | 1 | 3 | 5 | 1 | 3 | 5 | **1** | 3 | 5 |
| **TIC1** | 0.215 | 0.178 | 0.216 | 0.055 | 0.104 | 0.049 | 0.022 | 0.028 | 0.072 | **0.004** | 0.010 | 0.070 |
| **TIC2** | 0.191 | 0.154 | 0.167 | 0.049 | 0.110 | 0.069 | 0.023 | 0.031 | 0.099 | **0.004** | 0.009 | 0.065 |
| **RMSD** | 0.219 | 0.316 | 0.297 | 0.219 | 0.341 | 0.160 | 0.118 | 0.164 | 0.237 | **0.008** | 0.017 | 0.103 |
| **GDT** | 0.267 | 0.253 | 0.226 | 0.164 | 0.264 | 0.110 | 0.091 | 0.122 | 0.176 | **0.008** | 0.012 | 0.088 |
| **RG** | 0.257 | 0.302 | 0.132 | 0.208 | 0.347 | 0.173 | 0.141 | 0.171 | 0.252 | **0.025** | 0.033 | 0.162 |
| **FNC** | 0.281 | 0.374 | 0.192 | 0.129 | 0.160 | 0.082 | 0.071 | 0.077 | 0.142 | **0.003** | 0.011 | 0.111 |

Table 1: **Jensen-Shannon Divergence** of key observables from reference density. **TIC1** and **TIC2**: first two TICA components. **RMSD**: Root Mean Square Deviation of $\mathbf{C}_\alpha$ atoms to crystal reference. **GDT**: Global Distance Test (Total Score) of $\mathbf{C}_\alpha$ atoms to crystal reference. **RG**: Radius of Gyration. **FNC**: Fraction of Native Contacts. Please refer to Appendix A.7 for detailed metrics descriptions.

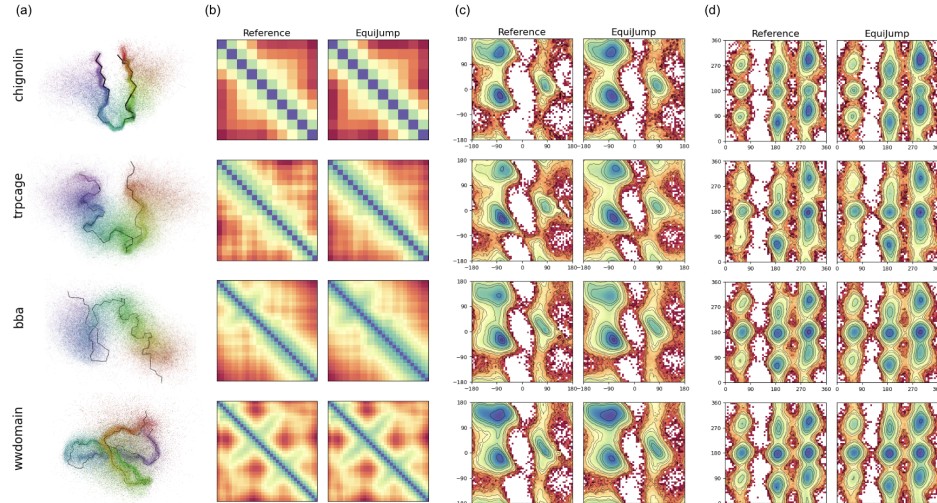

Figure 5: **EquiJump Samples**: **(a)** We visualize the distribution in 3D of 1500 backbone random samples of EquiJump trajectories. We align samples to the crystal backbone (shown in black) and verify that our model stays close to the native state basin. We show **(b)** mean pairwise $\mathbf{C}_\alpha$ distance matrices, **(c)** Ramachandran plots of backbone dihedrals and **(d)** Janin plots of sidechain dihedrals of EquiJump samples against reference trajectory data.

## 4.4 TRANSFERABLE MODEL

We investigate the capability and scaling performance of our method on the challenging task of learning stable and accurate dynamics for the 12 fast-folding proteins using a single transferable model. We systematically vary model capacity across $H = \{32, 64, 128, 256\}$ to assess the impact of model complexity on performance. We train all models for 500k steps with batch size 128. For collecting samples, we perform 500 simulations of 500 steps (50 ns) starting from states of the (enhanced) dataset. We employ 100 steps of integration to obtain the next configuration.

In Figure 5, we present samples of the largest EquiJump model and compare its generated densities of backbone and sidechain dihedral angles and pairwise $\mathbf{C}_\alpha$ distances to those of reference trajectory data. We provide plots for the additional fast-folding proteins in Appendix A.5. By accurately recovering distributional profiles across the sampled conformations, we demonstrate that EquiJump remains within the manifold of the original data. We further verify chemical validity by analyzing distributions of bond lengths and angles in Appendix A.6.

We compare the capabilities of our model across the different capacities and against available transferable model CG-MLFF (Majewski et al., 2023), which is based on coarse-graining and force matching, and uses Langevin sampling for producing dynamics. To the best of our knowledge, this is the only other multi-protein model that covers the 12 fast-folding proteins.

|  |  | **EquiJump** | | | |
|---|---|---|---|---|---|
|  | **CG-MLFF** | 32 | 64 | 128 | 256 |
| **TIC1** | 0.30 | 0.15 | 0.13 | 0.07 | **0.03** |
| **TIC2** | 0.23 | 0.17 | 0.09 | 0.06 | **0.03** |
| **RMSD** | 0.20 | 0.18 | 0.12 | 0.11 | **0.03** |
| **GDT** | 0.21 | 0.25 | 0.13 | 0.11 | **0.02** |
| **RG** | 0.18 | 0.14 | 0.08 | 0.12 | **0.04** |
| **FNC** | 0.27 | 0.25 | 0.13 | 0.08 | **0.03** |

Table 2: **Jensen-Shannon Divergence** of ensemble observables averaged over the 12 fast-folding proteins.

|  |  | **EquiJump** | | | |
|---|---|---|---|---|---|
|  | **CG-MLFF** | 32 | 64 | 128 | 256 |
| **RMSD** | 34.7 | 51.2 | 46.9 | 43.6 | **15.2** |
| **GDT** | 51.5 | 57.1 | 42.7 | 38.0 | **18.3** |
| **RG** | 9.4 | 13.8 | 11.4 | 18.7 | **4.3** |
| **FNC** | 45.2 | 48.8 | 32.8 | 23.7 | **15.7** |

Table 3: **Percent Error in Predicting Averages** of ensemble observables for the 12 fast-folding proteins

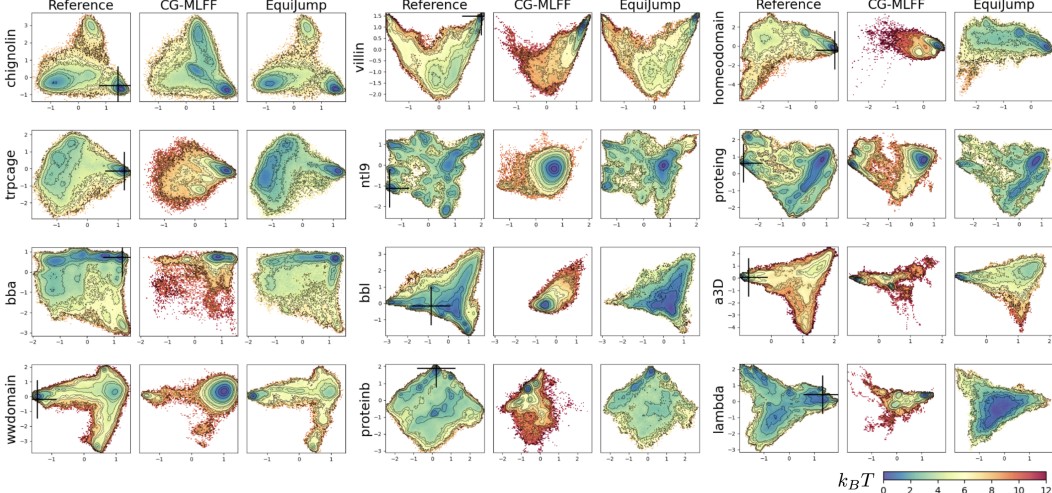

Figure 6: **Free Energy on TICA components for the 12 Fast-Folding Proteins**. We compare the free energy of EquiJump-256 against that of the reference and that of available model CG-MLFF. The free energy for each plot is set to 0 at the minimum and the color map is in units of $k_BT$ at the MD temperature of 350 K. EquiJump succesfully recovers the dynamics of the proteins, covering the phase space and stabilizing the basin of most (shown as + in reference profile).

In Tables 2 and 3, we investigate model performance in reproducing the long-time (MSM-reweighted, 4.2) distribution of ensemble observables and in recovering the average values of these observables. In both tables, metrics are averaged over the twelve proteins. We provide protein-specific results for Jensen-Shannon divergence comparisons in Appendix A.8. These results demonstrate that while the force field-based model is competitive in low-capacity regimes, our long-interval generative model significantly outperforms it in higher-capacity settings. This can be attributed to the fact that force-field methods are constrained to small time steps and only require short-term, local predictions which can be captured with fewer model parameters. In contrast, a model capable of large time steps must possess a deep understanding of the underlying data manifold, as the number of plausible transition states grows significantly with increasing time step size. While EquiJump requires substantial capacity to precisely reproduce equilibrium behavior, it successfully navigates the manifold across model sizes (Appendix A.9), while achieving overall superior performance.

For each model, we estimate the free energy profiles on the first two TIC dimensions across the 12 proteins and plot the distributions in Figure 6 and in Appendix A.9. In these plots, we observe that despite performing large steps of 100ps, large EquiJump models successfully describe the dynamics of the 12 fast-folding proteins by accurately recovering the free energy curves that describe long-term behavior. While CG-MLFF remains stable within most native basins, EquiJump covers a larger extension of the phase spaces and reveals stronger bias to less likely and disordered states, which is reduced with increasing model capacity (Appendix A.9). Ultimately, our model reveals better reconstruction of the slow components and more accurate profiling of the free energy TICA maps across the studied proteins. In Appendix A.10 we compare the free energy curves resulting from best performing EquiJump model and CG-MLFF on additional sample observables. Our findings indicate that large-scale generative transport models can outperform traditional neural force fields for long-term dynamics simulation.

### 4.4.1 PERFORMANCE ANALYSIS

In order to study the performance of our model, we consider the largest protein (lambda) as reference. The classical MD simulation used to generate its trajectory uses explicit water and the total system has size around 12000 atoms (Lindorff-Larsen et al., 2011). Following Amber24 benchmarks, on the same hardware we use for our simulations (NVIDIA A100) a system twice as large (JAC) can reach a throughput of 1258 ns/day (Exxact Corp., 2024). Scaling linearly to the size of the lambda cell, this results in 3.6s for a single 100ps step. Drawing from this reference, in Table 4 we compare the performance of EquiJump models across different scales, where we observe positive

|  | | Batch Size | | |
|---|---|---|---|---|
| Model (# Params) | | 1 | 8 | 32 |
| 32 (6.5M) | Time (s) | 0.34 | 1.49 | 3.35 |
| | Accel. ($\times$) | 10.60 | 19.33 | 34.39 |
| 64 (25.4M) | Time (s) | 0.51 | 2.26 | 6.07 |
| | Accel. ($\times$) | 7.05 | 12.74 | 18.98 |
| 128 (100.8M) | Time (s) | 0.60 | 3.12 | 12.17 |
| | Accel. ($\times$) | 6.00 | 9.23 | 9.47 |
| 256 (391.1M) | Time (s) | 1.05 | 6.40 | 26.09 |
| | Accel. ($\times$) | 3.40 | 4.50 | 4.42 |

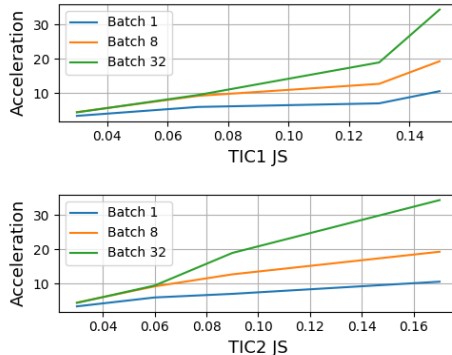

Table 4: **Performance Metrics and Estimates**. We measure the time of transport for a step of 100 ps using different model capacities when integrating with 100 latent time steps for simulating the largest protein considered (lambda), and estimate the acceleration factor from representative classical MD with explicit solvent that generated the training dataset. All results are reported on NVIDIA A100 machines with single GPU of 80G.

Figure 7: **Quality against Acceleration**. We plot estimated acceleration factors against Jensen-Shannon divergence (JS) for the distribution of long-term TIC components of Equi-Jump samples against reference trajectories. We display positive performance across batch sizes.

acceleration factors for all model instances. Based on these metrics, we study the relation of Equi-Jump speedup against generation quality. In Figure 7, we identify the trade-off between estimated acceleration factors and accuracy in reproducing the distribution of TIC components for different simulation batch sizes. We observe that EquiJump models are able to accurately reconstruct TIC components (JS $< 0.1$) while accelerating by factors of 5-15$\times$ compared to Amber24, achieving a significant simulation throughput with minimal trade-off in precision. In comparison, CG-MLFF is estimated to be 1-2 orders of magnitude slower than the reference simulation (Majewski et al., 2023). Similarly, state-of-the-art neural force-field MACE-OFF (Kovács et al., 2023) is estimated to perform 2.5Msteps/day for its smallest model on the same hardware (Kovács et al., 2023). With a time step of 4 fs, this corresponds to 860s per 100ps step, or a $0.004\times$ slowdown in comparison to reference. In contrast, despite its already promising acceleration, the performance of EquiJump is likely to be further enhanced through additional network architecture optimization, exploration of more efficient differential equation solvers, and application of distillation and sampling acceleration techniques (Luhman and Luhman, 2021; Salimans and Ho, 2022). In Appendix A.11, we study the impact of sampling hyperparameters and find promising results for more acceleration through further tuning of noise scaling for fewer integration steps.

## 5 CONCLUSION

In this work, we introduced EquiJump for learning the dynamics of 3D protein simulations. EquiJump extends Two-Sided Stochastic Interpolants for 3D dynamics through SO(3)-equivariant neural networks, and is implemented through a novel four-track architecture that handles all-atom structures. We validated our approach through a series of experiments on large-scale dataset of fast-folding proteins, through which we compared generative frameworks and demonstrated a unified model that accurately reproduces complex dynamics across different proteins. We ablated model capacities and compared our model to existing approaches, outperforming state-of-the-art methods in terms of accuracy and efficiency. Our results suggest EquiJump provides a stepping stone for future research in modeling and accelerating protein dynamics simulation. Future work will focus on architecture and parameter efficiency, and on transferability and generalization.

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

# A APPENDIX / SUPPLEMENTAL MATERIAL

## A.1 BROWNIAN DYNAMICS IN PROTEIN-SOLVENT SYSTEMS AND CONNECTION TO STOCHASTIC INTERPOLANTS

In the study of molecular dynamics of proteins immersed in solvents, it is crucial to account for the interactions between the proteins and the surrounding fluid. Proteins in a solvent experience random collisions with solvent molecules, leading to stochastic behavior that can be effectively modeled using Brownian dynamics (Ermak and McCammon, 1978). This approach captures the random motion arising from thermal fluctuations and solvent effects, providing a realistic depiction of protein behavior in biological environments.

Generalized frictional interactions among the particles can be incorporated into the Langevin equation through a friction tensor $\mathbf{R}$ (Schlick, 2010). This tensor accounts for the action of the solvent on the particles and modifies the Langevin equation to:

$$\mathbf{M}\ddot{\mathbf{X}}(t) = -\nabla E(\mathbf{X}(t)) - \mathbf{R}\dot{\mathbf{X}}(t) + \mathbf{W}(t), \tag{10}$$

where $\mathbf{M}$ is the mass matrix, $\mathbf{X}(t)$ represents the particle positions at time $t$, $E(\mathbf{X}(t))$ is the potential energy, and $\mathbf{W}(t)$ is a random force representing thermal fluctuations from the solvent. The mean and covariance of the random force $\mathbf{W}(t)$ are given by:

$$\langle \mathbf{W}(t) \rangle = 0, \quad \langle \mathbf{W}(t)\mathbf{W}(t')^T \rangle = 2k_B T \mathbf{R}\delta(t - t'), \tag{11}$$

where $k_B$ is Boltzmann's constant, $T$ is the temperature, and $\delta(t - t')$ is the Dirac delta function. This relation is based on the fluctuation-dissipation theorem, a fundamental result that connects the friction experienced by a particle to the fluctuations of the random force acting upon it, assuming the particle is undergoing random motion around thermal equilibrium.

This description ensures that the ensemble of trajectories generated from Eq. 10 is governed by the Fokker-Planck equation, a partial differential equation that describes the evolution of the probability density function of a particle's position and momentum in phase space during diffusive motion.

In this context, the random force $\mathbf{W}(t)$ is modeled as white noise with no intrinsic timescale. The inertial relaxation times, given by the inverses of the eigenvalues of the matrix $\mathbf{M}^{-1}\mathbf{R}$, define the characteristic timescale of the thermal motion. When these inertial relaxation times are short compared to the timescale of interest, it is appropriate to neglect inertia in the governing equation, effectively discarding the acceleration term by assuming $\mathbf{M}\ddot{\mathbf{X}}(t) \approx 0$. Under this approximation, Eq. 10 simplifies to the Brownian dynamics form:

$$\dot{\mathbf{X}}(t) = -\mathbf{R}^{-1}\nabla E(\mathbf{X}(t)) + \mathbf{R}^{-1}\mathbf{W}(t). \tag{12}$$

This simplification reflects that solvent effects are sufficiently strong to render inertial forces negligible, resulting in motion that is predominantly Brownian and stochastic in nature. This description is particularly effective for modeling very large, dense systems whose conformations in solution are continuously and significantly altered by the fluid flow in their environment.

To stably evolve Brownian dynamics eq. (12) over time, small integration steps are usually required due to the stiffness of the physical manifold. Molecular systems exhibit a wide range of timescales: fast atomic motions such as bond vibrations and thermal fluctuations occur on the order of femtoseconds, while slower conformational shifts and larger-scale rearrangements may take place over much longer periods. These necessitate small time steps to accurately capture the system's rapid changes without numerical instability. In contrast, Stochastic Interpolants eq. (9), while also following the form of eq. (12), enable smoothing of the data manifold by convolution with small Gaussian perturbations. This leads to a latent representation that is robust to noise, allowing for larger integration steps. The smoother manifold helps overcome local energy barriers and navigate the broader conformational landscape more efficiently, making it possible to simulate molecular dynamics on extended timescales without losing stability.

## A.2 Architecture Details

Algorithms 3, 4, 5 describe the components of EquiJump. The Tensor Square operation in Alg. 3 (line 1) is applied independently within each channel. The residual sum of Alg. 5 (line 5) is only performed on the geometric features, since positions are fixed.

Tested models employ irreps of `0e + 1e` across multiplicities $\{32, 64, 128, 256\}$. We only test conditional number of layers $L_{\text{cond}} = 6$, and header number of layers $L_{\text{header}} = 4$. Our experimentation indicates further scaling is a promising direction of research. For training the transferable model, we use $\sigma_{\mathbf{P}}^2 = 3.0$ and $\sigma_{\mathbf{V}}^2 = 1.0$.

For training all models we use the Adam optimizer (Kingma and Ba, 2017) with linearly decreasing learning rate from $1 \times 10^{-2}$ to $1 \times 10^{-3}$ over 150k steps. We perform all training experiments on NVIDIA A100 machines with 2-4 GPUs.

For interpolant parameterization we use $I(\tau, \mathbf{X}_0, \mathbf{X}_1) = (1 - \tau)\mathbf{X}_0 + \tau\mathbf{X}_1$, $\gamma(\tau) = \sigma \cdot \tau \cdot (1 - \tau)$ and fixed time dependent diffusion coefficient $\epsilon(\tau) = 1.0$ in sampling. Where $\sigma = 3.0, 1.0$ for the coordinates and geometric features, respectively. In future work we will investigate how different interpolant parameterizations affect the performance of our model.

---

**Algorithm 3** Self-Interaction

---

**Require:** Tensor Cloud $(\mathbf{P}, \mathbf{V})$
1: $\mathbf{V} \leftarrow \mathbf{V} \oplus (\mathbf{V})^{\otimes 2}$
2: $\mathbf{V} \leftarrow \text{MLP}(\mathbf{V}^{l=0}) \cdot \mathbf{V}$
3: $\mathbf{V} \leftarrow \text{Linear}(\mathbf{V})$
4: **return** $(\mathbf{P}, \mathbf{V})$

---

**Algorithm 4** Spatial Convolution

---

**Require:** Tensor Cloud $(\mathbf{V}, \mathbf{P})$
**Require:** Output Node Index $i$
1: $(\tilde{\mathbf{P}}, \tilde{\mathbf{V}})_{1:k} \leftarrow k\text{NN}(\mathbf{P}_i, \mathbf{P}_{1:N})$
2: $R_{1:k} \leftarrow \text{Embed}(||\tilde{\mathbf{P}}_{1:k} - \mathbf{P}_i||_2)$,
3: $\phi_{1:k} \leftarrow \text{SphericalHarmonics}(\tilde{\mathbf{P}}_{1:k} - \mathbf{P}_i)$
4: $\tilde{\mathbf{V}}_{1:k} \leftarrow \text{MLP}(R_k \oplus \mathbf{V}_{1:k}^{l=0} \oplus \mathbf{V}^{l=0}) \cdot \text{Linear}(\tilde{\mathbf{V}}_{1:k} \otimes \phi_{1:k})$
5: $\mathbf{V} \leftarrow \text{Linear}\left(\mathbf{V} + \frac{1}{k}\left(\sum_k \tilde{\mathbf{V}}_k\right)\right)$
6: **return** $(\mathbf{V}, \mathbf{P})$

---

**Algorithm 5** EquiJump Deep Network

---

**Require:** Tensor Cloud $\mathbf{X} = (\mathbf{P}, \mathbf{V}^{0:l_{\max}})$
1: $\mathbf{H}^0 \leftarrow \text{Self-Interaction}(\mathbf{X})$
2: **for** $l$ in [0, L) **do**
3: $\quad \mathbf{H}^{l+1} \leftarrow \text{Self-Interaction}(\mathbf{H}^l)$
4: $\quad \mathbf{H}^{l+1} \leftarrow \text{SpatialConvolution}(\mathbf{H}^{l+1})$
5: $\quad \mathbf{H}^{l+1} \leftarrow \text{LayerNorm}(\mathbf{H}^{l+1} + \mathbf{H}^l)$
6: $\mathbf{H}^{\text{agg}} \leftarrow \text{Linear}\left(\bigoplus_{l=0}^{L-1} \mathbf{H}^l\right)$
7: $\mathbf{H}^{\text{out}} \leftarrow \text{Self-Interaction}(\mathbf{H}^{\text{agg}})$
8: **return** $\mathbf{H}^{\text{out}}$

---

### A.2.1 Equivariance

The features used in EquiJump are irreducible representations (irreps) of SO(3). To prove that EquiJump is equivariant, we demonstrate that all operations within the network preserve the transformation properties of the irreps under rotation.

Scalars, which are irreps of degree $l = 0$, remain invariant under rotation. For a scalar $s \in \mathbb{R}$ and a rotation $R \in SO(3)$, we have:

$$R \cdot s = s. \tag{13}$$

Applying functions such as MLPs to scalars preserves this invariance:

$$f(R \cdot s) = f(s). \tag{14}$$

For irreps with $l > 0$, equivariance depends on the nature of the operations. Linear operations combine irreps of the same degree using scalar weights. Let $\mathbf{v}$ and $\mathbf{w}$ be irreps and $W$ a learnable weight matrix. Under a rotation $R$,

$$R \cdot (W\mathbf{v}) = W(R \cdot \mathbf{v}), \tag{15}$$

where $R \cdot \mathbf{v}$ represents the rotated vector. Since the weight matrix $W$ does not interfere with the transformation properties, linear operations are equivariant.

EquiJump also employs tensor products, which combine two irreps $\mathbf{v}$ and $\mathbf{w}$ of degrees $l_1$ and $l_2$, respectively, to produce new irreps of degrees $|l_1 - l_2|, \ldots, (l_1 + l_2)$. The tensor product transforms under rotation as:

$$R \cdot (\mathbf{v} \otimes \mathbf{w}) = (R \cdot \mathbf{v}) \otimes (R \cdot \mathbf{w}), \tag{16}$$

ensuring equivariance. In EquiJump, tensor products are used in two key cases: (1) between features and spherical harmonics $Y_{lm}(\mathbf{r})$ of relative positions, and (2) between features and themselves (tensor square). Spherical harmonics transform under rotation as:

$$R \cdot Y_{lm}(\mathbf{r}) = \sum_{m'} D^{(l)}_{mm'}(R) Y_{lm'}(\mathbf{r}), \tag{17}$$

where $D^{(l)}_{mm'}(R)$ are elements of the Wigner-D matrix. Combining features with spherical harmonics via tensor products preserves equivariance by construction.

The basic operation in EquiJump involves a tensor product followed by a linear transformation. Let $T$ represent this combined operation:

$$T(\mathbf{v}, \mathbf{w}) = W(\mathbf{v} \otimes \mathbf{w}), \tag{18}$$

where $W$ is a learnable weight tensor. Under a rotation $R$,

$$R \cdot T(\mathbf{v}, \mathbf{w}) = W(R \cdot (\mathbf{v} \otimes \mathbf{w})) = W((R \cdot \mathbf{v}) \otimes (R \cdot \mathbf{w})). \tag{19}$$

This demonstrates that this basic operation is equivariant. Since scalar transformations, linear layers, and tensor products all preserve equivariance, the entire EquiJump network is $SO(3)$-equivariant. This ensures that outputs transform consistently with inputs under rotation, making EquiJump well-suited for modeling the rotationally symmetric dynamics of protein structures in 3D space.

### A.3 DATASET DETAILS

We adapt the dataset produced by (Majewski et al., 2023). The dataset consists of trajectories of 500 steps at intervals of 100ps. Refer to the table below for the number of trajectories curated. To include all relevant residues, in addition to the standard vocabulary of residues, we also include a canonical form of Norleucine (NLE).

| Protein | Residues | Trajectories |
|---------|----------|--------------|
| Chignolin | 10 | 3744 |
| Trp-Cage | 20 | 3940 |
| BBA | 28 | 7297 |
| Villin | 34 | 17103 |
| WW domain | 35 | 2347 |
| NTL9 | 39 | 7651 |
| BBL | 47 | 18033 |
| Protein B | 47 | 6094 |
| Homeodomain | 54 | 1991 |
| Protein G | 56 | 11272 |
| a3D | 73 | 7113 |
| $\lambda$-repressor | 80 | 15697 |

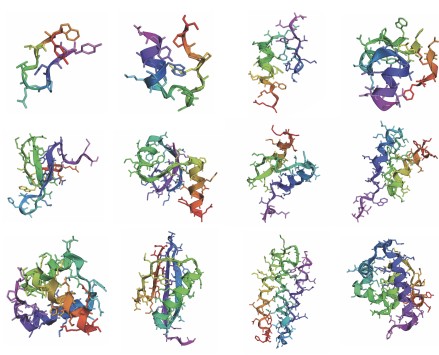

### A.4 DETAILS ON ENHANCED TRAINING CLUSTERS

To choose clusters that are diverse and dynamically relevant for enhanced sampling in training, we perform K-means clustering on 2D TICA components and find 200 clusters for each protein. Figure 8 visualizes cluster centers and the distribution of population sizes.

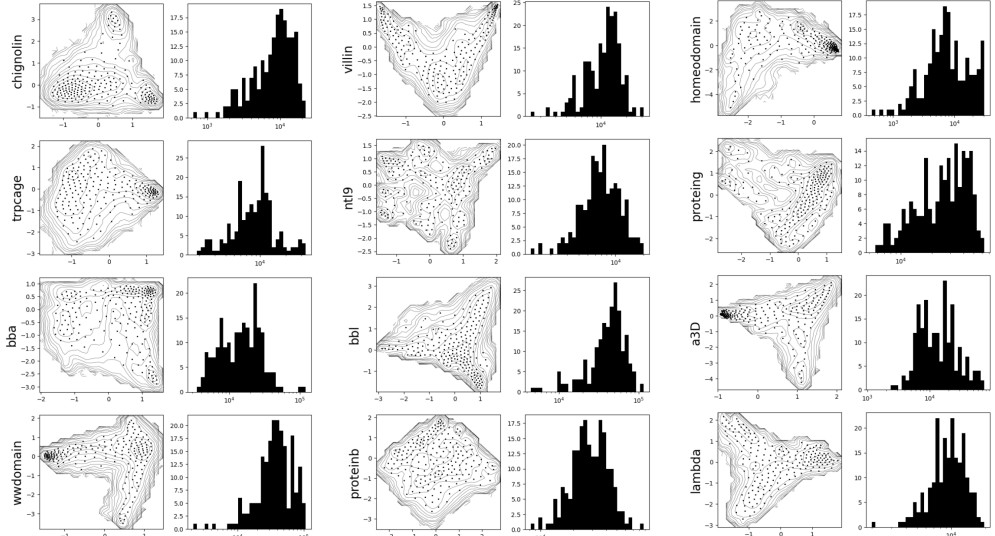

Figure 8: **Cluster Centers and Distribution of Cluster Population Sizes**.

A.5 ADDITIONAL VISUALIZATION OF SAMPLES

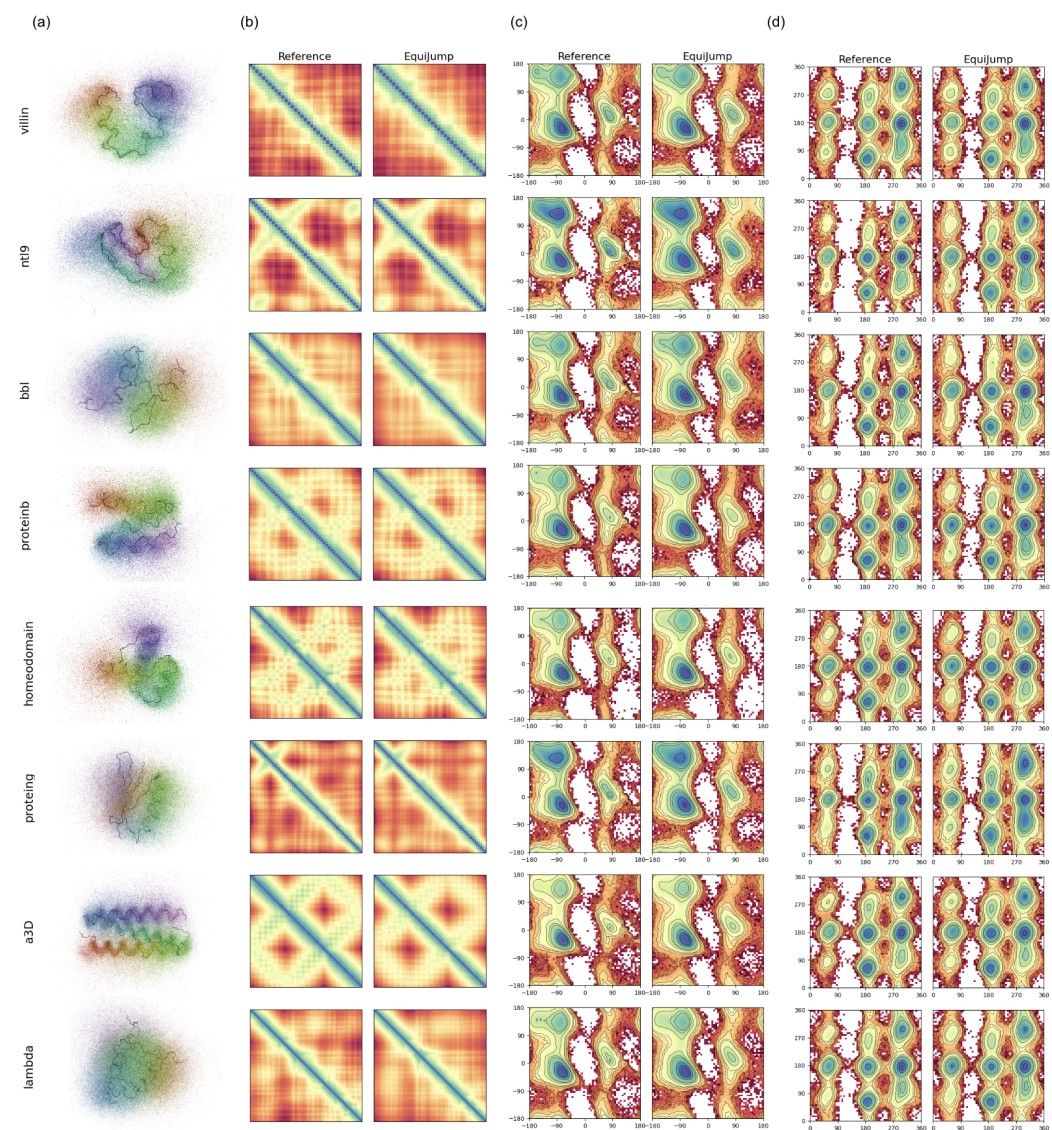

Figure 9: **EquiJump Samples**: **(a)** We visualize the performance of EquiJump on additional fast-folding proteins by superposing 1500 backbone random samples of EquiJump trajectories. We align samples to the crystal backbone (shown in black). We further present **(b)** mean pairwise $C_\alpha$ distance matrices, **(c)** Ramachandran plots of backbone dihedrals and **(d)** Janin plots of sidechain dihedrals of EquiJump samples against reference trajectory data.

## A.6 CHEMISTRY EVALUATION

We verify that EquiJump trajectories stay in a chemically valid manifold by plotting the distribution of bond lengths, bond angles and collisions of van der Waals radii against those in reference. In Figure 10, we plot those distributions for best peforming EquiJump model ($H = 256$) for fast-folding proteins Chignolin, Trp-cage, BBA and the WW domain. We observe that EquiJump accurately reproduces the distribution curves, interestingly revealing fewer counts for atomic clashes (despite larger dispersion) compared to reference. Our plots demonstrate that our model produces data that successfully stays within a chemically valid manifold across the different proteins.

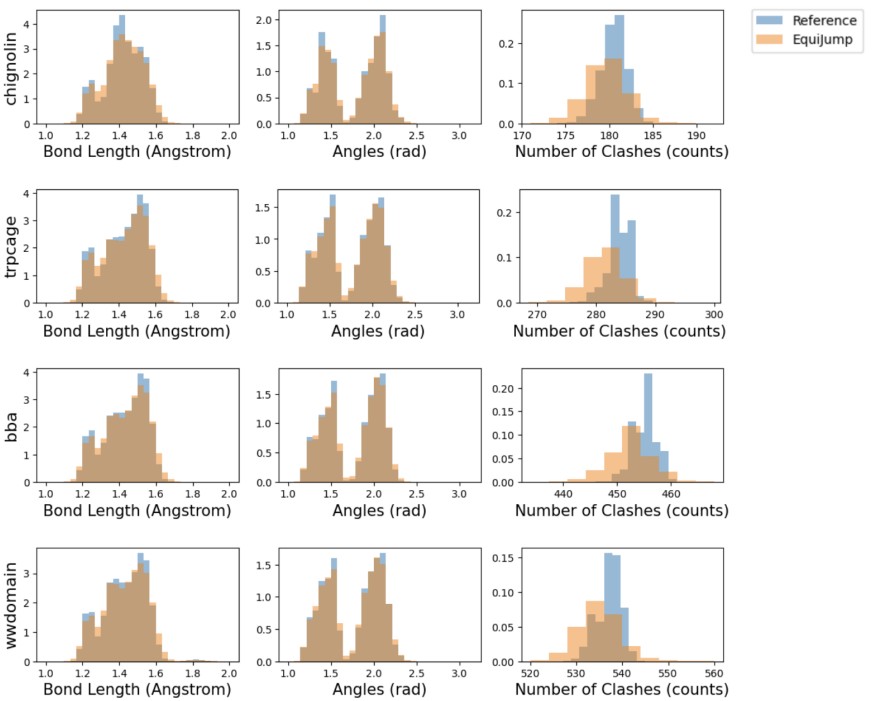

Figure 10: **Distribution of Chemical Measures**. We pick 3000 structures at random from reference trajectories and EquiJump. For each sample, we measure the distribution of bond lengths and bond angles considering all heavy atoms in the system. We estimate clashes by counting intersections of van der Waals radii for all pairs of non-bonded atoms.

A.7 METRICS

In partiular we use the following metrics for assessment as commonly used for protein dynamics analysis:

**TICs (Time-lagged Independent Components):** Derived from Time-lagged Independent Component Analysis (TICA), TICs project molecular dynamics data into a reduced-dimensional space that emphasizes slow collective motions (Pérez-Hernández et al., 2013). We fit TICA on ground truth and project samples to make comparisons. We consider the two main TIC components for our metrics. For embedding our proteins for TICA, we use a distance matrix considering only $C_\alpha$ positions. We use lagtime of 2ns.

**RMSD (Root Mean Square Deviation):** RMSD measures the average deviation of atomic positions from a reference structure:

$$\text{RMSD} = \sqrt{\frac{1}{N}\sum_{i=1}^{N}\|\mathbf{x}_i^{\text{model}} - \mathbf{x}_i^{\text{ref}}\|^2}, \tag{20}$$

where $N$ is the number of atoms. We align $\mathbf{C}_\alpha$ backbone samples to reference crystal structure and measure the resulting RMSD for obtaining our metric.

**GDT (Global Distance Test):** GDT quantifies the fraction of residues within specific distance thresholds of the reference structure:

$$\text{GDT} = \frac{1}{4}\sum_{d\in\{1,2,4,8\}}\frac{\text{\# of residues within } d\,\text{Å}}{\text{total residues}}, \tag{21}$$

and is used in protein structure alignment (Zemla, 2003). We align the $\mathbf{C}_\alpha$ of our structures to reference crystal structure and measure GDT for obtaining our reported values.

**RG (Radius of Gyration):** The compactness of a protein structure is measured as:

$$R_g = \sqrt{\frac{\sum_{i=1}^{N} m_i\|\mathbf{x}_i - \mathbf{x}_{\text{COM}}\|^2}{\sum_{i=1}^{N} m_i}}, \tag{22}$$

where $m_i$ and $\mathbf{x}_i$ are the mass and position of atom $i$, and $\mathbf{x}_{\text{COM}}$ is the center of mass.

**FNC (Fraction of Native Contacts):** FNC evaluates the fraction of interatomic contacts preserved from the reference structure:

$$\text{FNC} = \frac{\text{\# of native contacts in the sample}}{\text{\# of native contacts in the reference crystal}}. \tag{23}$$

This metric highlights native structure preservation (Best et al., 2013).

## A.8 PROTEIN-SPECIFIC ABLATION RESULTS

We report Jensen-Shannon Divergence (Lin, 1991) against reference trajectories for reweighted ensemble observables of each protein, comparing CG-MLFF (Majewski et al., 2023) and EquiJump at various capacities to assess their ability to replicate realistic structural and dynamic properties.

| | | CG-MLFF | EquiJump | | | |
|---|---|---|---|---|---|---|
| | | | 32 | 64 | 128 | 256 |
| chignolin | TIC1 | 0.221 | 0.026 | 0.069 | 0.009 | **0.006** |
| | TIC2 | 0.152 | 0.019 | 0.039 | **0.003** | 0.006 |
| | RMSD | 0.253 | 0.028 | 0.083 | **0.012** | 0.018 |
| | GDT | 0.231 | 0.018 | 0.080 | **0.010** | 0.013 |
| | RG | 0.191 | 0.029 | 0.061 | **0.008** | 0.020 |
| | FNC | 0.189 | 0.024 | 0.069 | **0.007** | 0.012 |
| trpcage | TIC1 | 0.372 | 0.115 | 0.107 | 0.048 | **0.019** |
| | TIC2 | 0.187 | 0.037 | 0.047 | 0.068 | **0.008** |
| | RMSD | 0.283 | 0.051 | 0.038 | **0.011** | 0.022 |
| | GDT | 0.302 | 0.066 | 0.042 | **0.013** | 0.021 |
| | RG | 0.438 | 0.045 | 0.028 | **0.007** | 0.035 |
| | FNC | 0.299 | 0.100 | 0.096 | 0.036 | **0.031** |
| bba | TIC1 | 0.334 | 0.110 | 0.067 | 0.114 | **0.044** |
| | TIC2 | 0.169 | 0.132 | **0.012** | 0.022 | 0.017 |
| | RMSD | **0.022** | 0.102 | 0.048 | 0.211 | 0.025 |
| | GDT | 0.037 | 0.339 | 0.045 | 0.248 | **0.029** |
| | RG | 0.185 | 0.086 | **0.024** | 0.271 | 0.026 |
| | FNC | 0.200 | 0.279 | 0.086 | 0.143 | **0.026** |
| wwdomain | TIC1 | 0.252 | 0.191 | 0.061 | 0.048 | **0.028** |
| | TIC2 | 0.072 | 0.065 | 0.047 | 0.037 | **0.014** |
| | RMSD | 0.246 | 0.226 | 0.091 | 0.139 | **0.022** |
| | GDT | 0.263 | 0.240 | 0.093 | 0.127 | **0.021** |
| | RG | 0.084 | 0.161 | 0.064 | 0.173 | **0.018** |
| | FNC | 0.264 | 0.294 | 0.100 | 0.090 | **0.021** |
| villin | TIC1 | 0.347 | 0.181 | 0.091 | 0.020 | **0.015** |
| | TIC2 | 0.340 | 0.162 | 0.078 | **0.016** | 0.019 |
| | RMSD | 0.253 | 0.149 | 0.079 | 0.035 | **0.016** |
| | GDT | 0.293 | 0.151 | 0.088 | 0.028 | **0.015** |
| | RG | 0.240 | 0.101 | 0.054 | 0.079 | **0.019** |
| | FNC | 0.300 | 0.149 | 0.064 | **0.015** | 0.020 |
| ntl9 | TIC1 | 0.251 | 0.270 | 0.207 | 0.072 | **0.045** |
| | TIC2 | 0.270 | 0.287 | 0.225 | 0.078 | **0.073** |
| | RMSD | 0.192 | 0.283 | 0.191 | 0.101 | **0.059** |
| | GDT | 0.170 | 0.242 | 0.156 | 0.069 | **0.044** |
| | RG | **0.019** | 0.187 | 0.130 | 0.121 | 0.050 |
| | FNC | 0.172 | 0.383 | 0.240 | 0.054 | **0.038** |
| bbl | TIC1 | 0.402 | 0.124 | 0.062 | 0.069 | **0.033** |
| | TIC2 | 0.229 | 0.224 | 0.063 | 0.137 | **0.036** |
| | RMSD | 0.378 | 0.053 | 0.016 | 0.135 | **0.011** |
| | GDT | 0.409 | 0.055 | **0.008** | 0.132 | **0.008** |
| | RG | 0.207 | 0.042 | **0.013** | 0.140 | 0.018 |
| | FNC | 0.445 | 0.237 | 0.057 | 0.134 | **0.029** |
| proteinb | TIC1 | 0.377 | 0.055 | 0.040 | 0.041 | **0.008** |
| | TIC2 | 0.332 | 0.115 | 0.062 | 0.054 | **0.008** |
| | RMSD | 0.214 | 0.265 | 0.156 | 0.178 | **0.007** |
| | GDT | 0.240 | 0.277 | 0.162 | 0.181 | **0.007** |

|  |  | CG-MLFF | EquiJump | | | |
|---|---|---|---|---|---|---|
|  |  |  | 32 | 64 | 128 | 256 |
|  | RG | 0.247 | 0.247 | 0.121 | 0.190 | **0.044** |
|  | FNC | 0.313 | 0.189 | 0.095 | 0.095 | **0.004** |
| homeodomain | TIC1 | 0.250 | 0.308 | 0.260 | 0.203 | **0.081** |
|  | TIC2 | 0.183 | 0.144 | 0.130 | 0.117 | **0.044** |
|  | RMSD | 0.150 | 0.414 | 0.298 | 0.321 | **0.068** |
|  | GDT | 0.189 | 0.468 | 0.349 | 0.355 | **0.078** |
|  | RG | **0.051** | 0.288 | 0.195 | 0.313 | 0.137 |
|  | FNC | 0.246 | 0.378 | 0.257 | 0.261 | **0.089** |
| proteing | TIC1 | 0.180 | 0.077 | 0.127 | 0.034 | **0.009** |
|  | TIC2 | 0.212 | 0.602 | 0.154 | 0.037 | **0.012** |
|  | RMSD | 0.075 | 0.175 | 0.101 | 0.079 | **0.019** |
|  | GDT | 0.103 | 0.628 | 0.106 | 0.067 | **0.013** |
|  | RG | 0.079 | 0.191 | 0.069 | 0.105 | **0.040** |
|  | FNC | 0.241 | 0.386 | 0.155 | 0.039 | **0.011** |
| a3d | TIC1 | 0.348 | 0.336 | 0.356 | 0.095 | **0.072** |
|  | TIC2 | 0.319 | 0.130 | 0.099 | 0.055 | **0.034** |
|  | RMSD | 0.107 | 0.352 | 0.336 | 0.074 | **0.070** |
|  | GDT | 0.112 | 0.355 | 0.339 | 0.072 | **0.057** |
|  | RG | 0.371 | 0.234 | 0.173 | 0.099 | **0.038** |
|  | FNC | 0.224 | 0.311 | 0.286 | 0.079 | **0.055** |
| lambda | TIC1 | 0.330 | 0.109 | 0.159 | **0.107** | 0.116 |
|  | TIC2 | 0.338 | 0.210 | 0.144 | 0.100 | **0.091** |
|  | RMSD | 0.311 | 0.129 | 0.109 | **0.033** | 0.046 |
|  | GDT | 0.277 | 0.167 | 0.095 | **0.028** | 0.042 |
|  | RG | 0.157 | 0.137 | 0.131 | **0.046** | 0.053 |
|  | FNC | 0.382 | 0.277 | 0.116 | **0.044** | 0.060 |

## A.9 ADDITIONAL TICA FREE ENERGY PROFILES

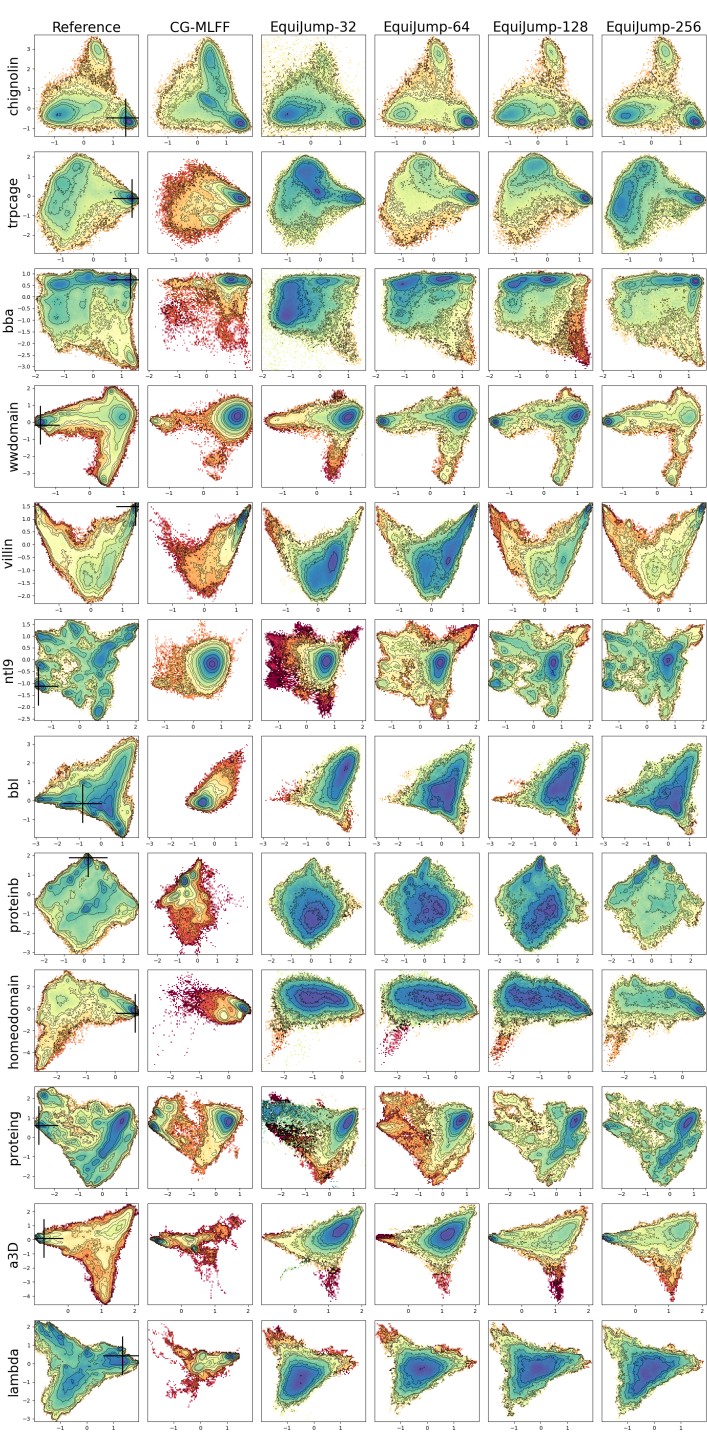

Figure 11: **From disorder to order: Free Energy profiles on TIC1 and TIC2 for comparison model and EquiJump models with increasing capacity.** While the MLFF model remains close to basin states, EquiJump is biased to less ordered regions despite staying in the manifold, and instead becomes more stable with increasing capacity.

## A.10 ADDITIONAL FREE ENERGY CURVES

In Figure 12, we show the estimated free energy of observable $C_\alpha$ Root Mean Square Deviation (RMSD) from the reference crystal structure, following reweighting based on stationary distributions of fitted Markov Models. We compare the best performing EquiJump model against reference and CG-MLFF. These curves represent the energy distribution of $C\alpha$-RMSD after the system reaches equilibrium and are highly sensitive to the accurate estimation of conformational transitions, making them a robust evaluation metric for dynamics models. We additionally provide free energy curve estimates for Radius of Gyration 13 and for Fraction of Native Contacts 14.

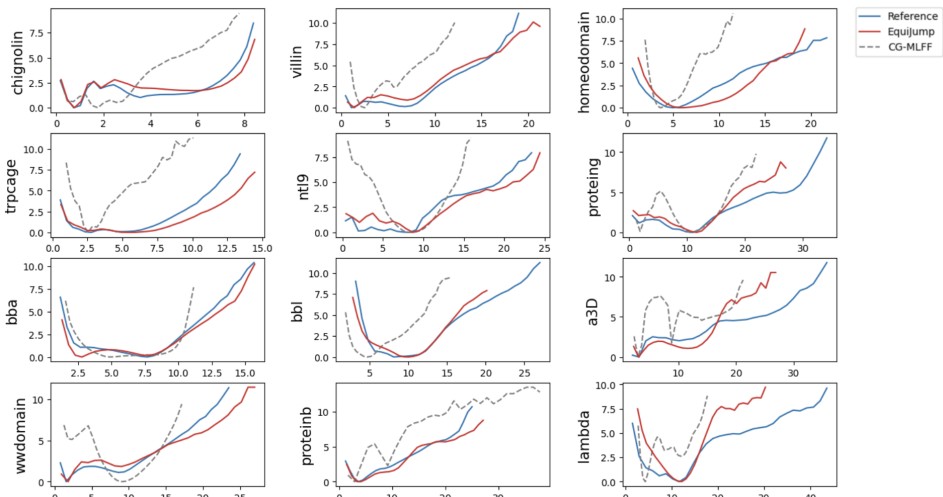

Figure 12: **Free Energy on $C_\alpha$-RMSD for the 12 Fast-Folding Proteins**. We align trajectory samples to the reference crystal, and measure $C_\alpha$-RMSDs (x-axis). Using Markov State Model (MSM) weights based on our TICA-based clusters, we reweight $C_\alpha$-RMSD counts to obtain free energy estimates (y-axis). We find that EquiJump successfully approximates the free energy curves of reference trajectories.

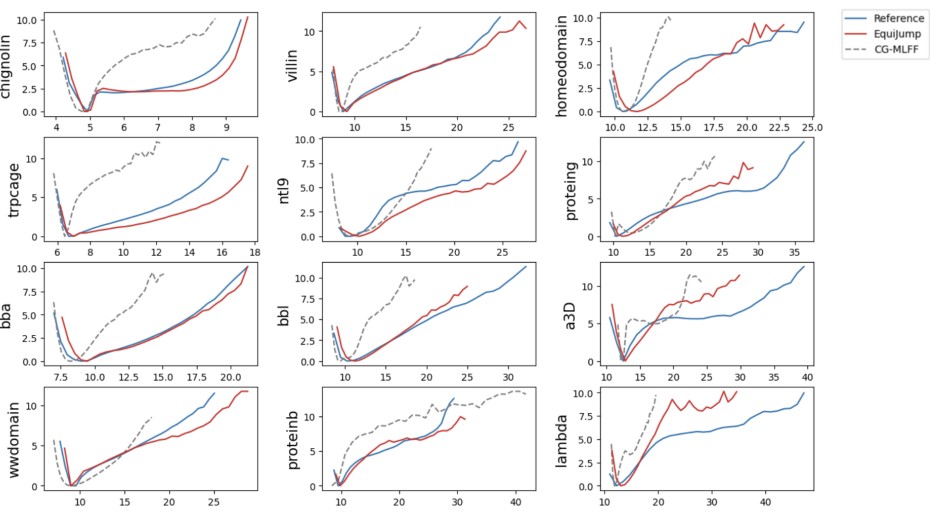

Figure 13: **Free Energy on $C_\alpha$ Radius of Gyration for the 12 Fast-Folding Proteins**. We bin and reweigh counts of $C_\alpha$ gyradii (x-axis) based on MSM weights to obtain free energy estimates (y-axis).

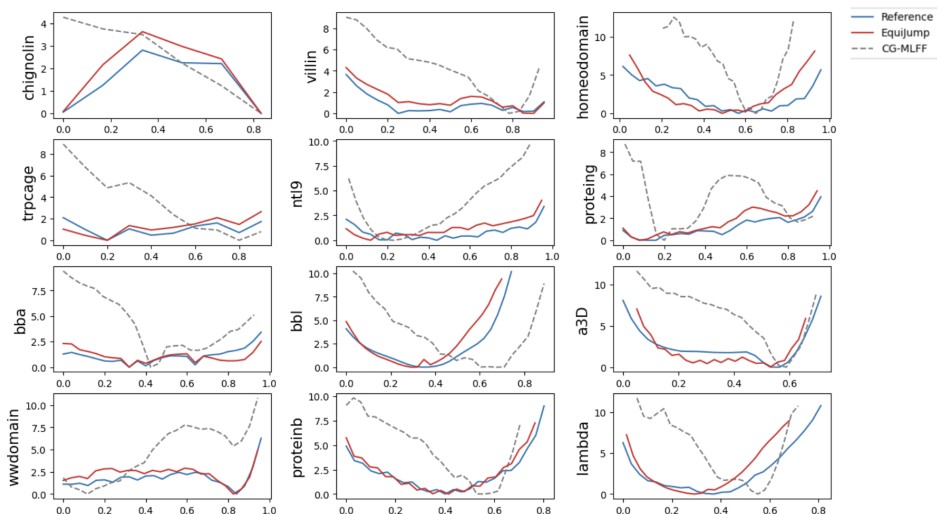

Figure 14: **Free Energy on Fraction of Native Contacts of** $C_\alpha$ **atoms for the 12 Fast-Folding Proteins**. We bin and reweigh Fraction of Native Contacts (FNC) (x-axis) of $C_\alpha$ atoms based on MSM weights to obtain free energy estimates (y-axis). We only consider residues at least 3 sequence positions apart, and use a cutoff of 8 Å for counting contacts.

## A.11 SAMPLING PARAMETERS ABLATION

We study the impact of changing sampling hyperparameters $\epsilon(\tau)$ and $d\tau$ in the quality of long-term dynamics generation. For that, we train a model for simulating the dynamics of fast-folding protein Chignolin. We parameterize our model with $H = 32$ and train it for $120k$ steps with batch size 512. We consider scale of noising parameter $\epsilon(\tau) = \{0.1, 0.3, 0.5, 1.0, 2.0, 3.0, 10.0\}$ and number of steps in integration $(1/d\tau) = \{30, 50, 75, 100, 150, 200, 300\}$. For each, we sample 300 trajectories of 500 steps (50 ns). As above, we reweight metrics through MSM based on TICA-clusters to estimate values at equilibrium.

In Figure 15, we show the Jensen-Shannon Divergence (JS) between ground truth and samples obtained at different parameterizations of $\epsilon$ and $d\tau$. In Figure 16, we show the effects of hyperparamter variation on the (long-term reweighted) density of the first TIC components of samples. We observe that large variation ($= 0.1, 10.0$) on $\epsilon$ yields underperforming models. Notably, we observe that even with few steps ($30, 50, 75$) higher quality can be obtained through smaller $\epsilon$ ($= 0.5, 0.75$). Our results suggest that further investigation about the noising schedule $\epsilon(\tau)$ is a promising direction for increasing acceleration of high-quality simulation through fewer integration steps.

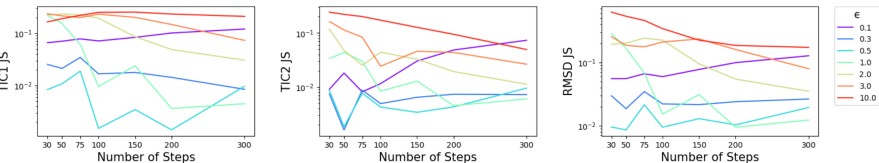

Figure 15: **Effect of Sampling Parameters on Generation Quality**. We measure Jensen-Shannon divergence (JS) from reference of (reweighted) observable distributions (TIC1, TIC2 and RMSD) across different sampling parameters $\epsilon(\tau)$ and $(1/d\tau)$.

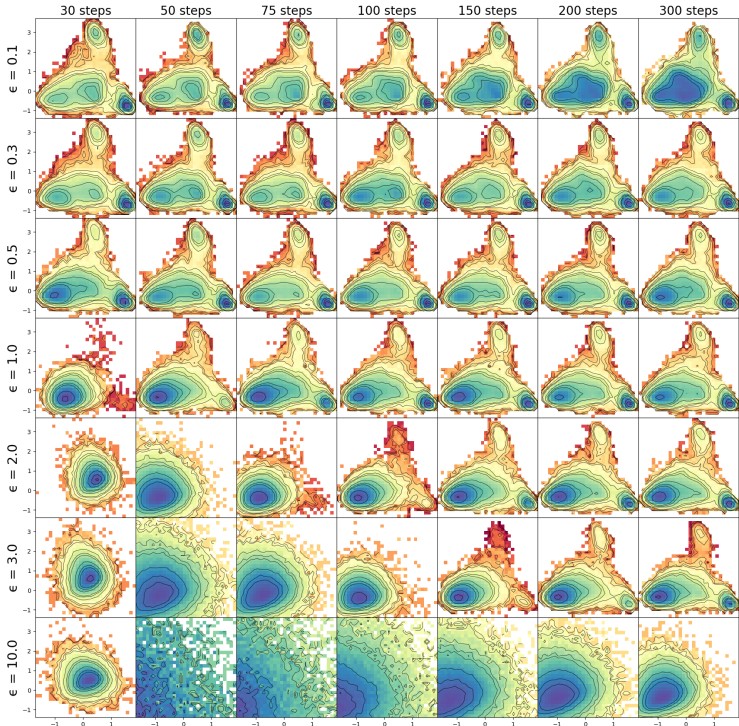

Figure 16: **Effect of Sampling Parameters on TIC profile**: we plot the density of the first TIC components of Chignolin with varying noise factor $\epsilon(\tau)$ and integration number of steps $(1/d\tau)$.

