# OpenReview forum: "EquiJump: Protein Dynamics Simulation via SO(3)-Equivariant Stochastic Interpolants"
_ICLR.cc/2025/Conference — Submitted to ICLR 2025_

### Official Review · Reviewer_zvV7 · 2024-10-22

**Soundness:** 3
**Presentation:** 4
**Contribution:** 2
**Rating:** 5
**Confidence:** 4

**Summary:**

This work focuses on enhancing molecular dynamics of proteins and proposes EquiJump, a two-sided stochastic interpolant framework tailored for arbitrary data endpoints with $\mathrm{SO}(3)$-equivarient modules. Experiments on 12 fast-folding proteins verify its superiority by accurately replicating the dynamics.

**Strengths:**

1. A good try to leverage two-sided stochastic interpolants for MD, which can be considered as a Markov process and therefore serves as a suitable scenario for bridging the arbitrary data endpoints.
2. The work proposes the tensor cloud representation and a corresponding $\mathrm{SO}(3)$-equivariant module, obtaining a more detailed representation of 3D coordinates.
3. Inspired by classical MD methods like umbrella sampling, the work utilizes the resampling technique to uniformly sample different states from MD trajectories for training.

**Weaknesses:**

1. This work is slightly lacking in innovation: the concept and conclusions of the two-sided stochastic interpolant were elucidated in [1], and the $SO(3)$-equivariant architecture of EquiJump is similar to other existing architectures as well, such as Equiformer [2]. Further explanations of the neccesity to use such architectures are warranted.
2. EquiJump was trained and evaluated both on the same dataset (fast-folding proteins). Although the dynamics are restored accurately, there's no proof for the transferability to other protein systems of the model.
3. EquiJump was only compared to CG-MLFF in this work, where CG-MLFF is performed in a corase-grained fashion, leading to a relatively unfair comparison. It would be more convicing to compare with more advanced models that work on all-atom systems, such as Timewarp [3]. Meanwhile, providing statistical results will be better.

**Reference**

[1] Albergo, M. S., Boffi, N. M., & Vanden-Eijnden, E. (2023). Stochastic interpolants: A unifying framework for flows and diffusions. arXiv preprint arXiv:2303.08797.

[2] Liao, Y. L., & Smidt, T. (2022). Equiformer: Equivariant graph attention transformer for 3d atomistic graphs. arXiv preprint arXiv:2206.11990.

[3] Klein, L., Foong, A., Fjelde, T., Mlodozeniec, B., Brockschmidt, M., Nowozin, S., ... & Tomioka, R. (2024). Timewarp: Transferable acceleration of molecular dynamics by learning time-coarsened dynamics. Advances in Neural Information Processing Systems, 36.

**Questions:**

1. Please give more explanations of the neccesity to propose a novel $\mathrm{SO}(3)$-equivariant architecture rather than using existing architecutres like Equiformer. Ablation results will be better.
2. To prepare the training dataset, the resampling technique was used to uniformly sample states with different free energies. I wonder that while the dataset was biased from the Boltzmann distribution, the minimizer of the training objective might also be biased from the real dynamics. Please correct me if I am wrong.
3. Please provide more experimental results of stronger baselines by training from scratch if possible. It would be more appreciated if the results could be reported through statistical metrics.

**Details Of Ethics Concerns:**

No concerns.

---

> ### Author Response · Authors · 2024-11-21
>
> We thank the reviewer for their questions and comments.
>
> We are working to clarify the points raised by the reviewer, here are some comments on the remarks:
> - While stochastic interpolants and equivariant graph neural networks (GNNs) are established concepts, our work is the first to combine them for modeling long-timescale protein dynamics at the atomic level, bridging the gap between conception and practice. This required addressing challenges such as efficient protein representation, resampling training data of molecular states, and ensuring accurate thermodynamic properties. The integration of stochastic interpolants with an SO(3)-equivariant architecture goes beyond existing frameworks by enabling efficient, long-timestep probabilistic dynamics and comprehensive phase space exploration. We provide significant speed-ups while accurately reproducing free energy landscapes and dynamical distributions for 12 fast-folding proteins, enabling a scalable and practical solution for molecular dynamics. We will further clarify these contributions in the revised manuscript..
> - We only describe the model as transferable across the 12 proteins of the training set, and do not expect it to be able to generalize securely to unseen data. Here, we follow the use of the literature for the term “transferable”, and keep “generalizable” for testing on out-of-distribution. It is true that our model is not tested for transferability to other proteins. However, compared with the literature, the capacity of our model to correctly reproduce the dynamics of these 12 fast folding proteins surpasses any other attempts so far (CGMLFF) both in accuracy and scope. Only the current work and CGMLFF use a single model for different proteins (and we show we can better mimic the dynamics on multiple proteins at once), while other works train separate models for different proteins and thus inherently lack transferability. We believe that with appropriate scaling, this work could offer a promising stepping stone towards a truly generalized model for protein dynamics.

---

> ### Author Response · Authors · 2024-11-21
>
> - While Timewarp does enable large scale time evolution as EquiJump, the authors only focus on small scale (2-4 amino acids) molecular systems, and it is unclear whether the model would be directly comparable. However, we are working on providing a generative comparison of the Two-Sided interpolants against Flow Matching and DDPM, which share similar underlying principles to the method used in Timewarp (which generates next frames from a source Gaussian distribution). We share preliminary results below.
> > The tables below make comparisons across several metrics. These are:
> > - **TICS**: Time-lagged Independent Component Analysis (TICA) is used to project molecular dynamics into a reduced dimensional space, emphasizing slow collective motions.
> > - **RMSD (Root Mean Square Deviation)**: Measures the average deviation of atomic positions from a reference structure.
> > - **GDT-TS (Global Distance Test - Total Score)**: A structural alignment metric evaluating the proportion of atoms in the model trajectory that are within a specified distance threshold of the reference structure.
> > - **RG (Radius of Gyration)**: Describes the compactness of the protein structure, measuring the spread of atoms around the protein's center of mass.
> > - **FNC (Fraction of Native Contacts)**: Fraction of interatomic contacts in the model trajectory that match those in the reference structure.
> >
> > Jensen-Shannon Divergence (Results on Protein G):
> > | Metric | DDPM | Flow | EquiJump |
> > |---|---|---|---|
> > | TIC1 | 0.215 | 0.055 | **0.004**
> > | TIC2 | 0.191 | 0.049 | **0.004**
> > | RMSD | 0.219 | 0.219 | **0.008**
> > | GDT-TS | 0.267 | 0.164 | **0.008**
> > | RG | 0.257 | 0.208 | **0.025**
> > | FNC | 0.281 | 0.129 | **0.003**
> >
> > We also include further statistical metrics for comparing between CG-MLFF and EquiJump:
> >
> >	Jensen-Shannon Divergence:
> > | Metric | CG-MLFF | EquiJump |
> > |---|---|---|
> > | TIC1 | 0.30 | **0.03** |
> > | TIC2 | 0.23 | **0.03** |
> > | RMSD | 0.20 | **0.03** |
> > | GDT-TS | 0.21 | **0.02** |
> > | RG | 0.18 | **0.04** |
> > | FNC | 0.27 | **0.03** |
> >
> >Percent Error of Mean Observable Value:
> > | Metric | CG-MLFF | EquiJump |
> > |---|---|---|
> > | RMSD | 34.7 | **15.2** |
> > | GDT-TS | 57.1 | **18.3** |
> > | RG | 9.4 | **4.3** |
> > | FNC | 45.2 | **15.7** |
>
> And answers to the specific questions:
>
> 1. We have no specific reason to prefer our architecture over other existing equivariant GNN architectures. Indeed, the network we employ for each head is not very different from existing ones (e.g. Tensor Field Networks, NequIP) and it would be possible to use Equiformer, once adapted to be used within our stochastic interpolants framework. We are updating the text to emphasize these parallels. The primary focus of this work lies in applying the stochastic interpolants framework for learning 3D molecular dynamics rather than introducing a fundamentally new equivariant network architecture. The choice to develop our own implementation stems from the need to integrate with this framework and ensure consistency in evaluating its potential for dynamics learning. Future work could explore alternative architectures like Equiformer or others to assess their efficacy within this framework.
> 2. While we do reweigh the samples in our dataset, we are careful to only choose different starting points, while always using the corresponding next step as a target. This implies that our network learns from different parts of the phase space with different weights, but for any initial point the output is sampled from the original Boltzmann distribution. Thanks to this, we ensure that the dynamics itself, given a starting point, will not be ultimately biased.
> 3. We have now included more tables with statistical evaluation of our model compared with the ground truth and the CGMLFF model.

---

> > ### Comment · Reviewer_zvV7 · 2024-11-22
> >
> > Thanks to the author's response and I appreciate the efforts to introduce those widely-used metrics for better comparisons. However, there are still some concerns that have not been addressed, which are listed below.
> >
> > - I still believe that validating the model's transferability on OOD (out-of-distribution) data is essential. In practical applications, we cannot obtain dynamic data for an unfamiliar molecule in advance, so only models with a certain level of transferability will be useful. That's also one of the reasons for applying deep learning methods in molecular dynamics.
> > - I suggest using related open-source models as baselines rather than the self-implemented flow model, as the latter lacks persuasiveness.
> > - I think the author did not fully understand the impact of resampling on the learned distributions of the model, which I mentioned in Question 2. The authors explained that despite using resampling techniques, they always used the corresponding next step as a target. This can be expressed as $\rho(x_{t+1}|x)\delta(x-x_t)$ remaining unchanged, where $\delta(\cdot)$ is the dirac mass. However, the two-sided interpolants actually fits **the following expectation** $E_{x_t}\rho(x_{t+1}|x)\delta(x-x_t)$. While the distribution of $x_t$ is biased from the Boltzmann distribution by resampling, the minimizer will also be biased. This is exactly the issue I want to discuss.
> >
> > In summary, I will maintain my score.

---

> ### Author Response · Authors · 2024-11-29
>
> We thank the reviewer for their thoughtful comments and would like to clarify our work and update the questions with new baselines and analyses:
> - We agree that our ultimate modeling goal for usefulness is to have a transferable model that can generalize to unseen proteins. While EquiJump is not explicitly tested for out-of-distribution (OOD) generalization, we extensively experiment with it on the difficult task of accurately fully fitting the long-term dynamics of 12 proteins while performing large steps. The considered dataset includes millions of conformational states and transitions, and we focus our attention on the task of learning models that reproduce long-term dynamics with precision, which we found to be a challenging task. While these experiments are still not geared towards OOD performance, we consider them important stepping stones to future work in the regime of larger models and larger datasets for investigating generalization. We hope EquiJump can provide a solid study for this direction.
> - Compared to methods like Timewarp, which focus on small systems (2–4 residues) and short-term dynamics through Normalizing Flows [1], EquiJump focuses on long-term dynamics of large proteins and uses more recent model of Stochastic Interpolants [2]. Unfortunately, it is hard to directly compare Timewarp and our approach, as they are studied on different datasets and system scales. Specifically, Timewarp as available does not scale beyond small systems and lacks training weights that would enable direct comparison to the scales we study; similarly EquiJump does not have the architectural design needed for Normalizing Flows (e.g. invertibility). Instead, to address this limitation, in Section 4.3 we compare our model to Flow Matching [3], which shares similar underlying principles to the generative method of Timewarp. We additionally compare our model to other state-of-the-art generative models, including Denoising Diffusion Probabilistic Models (DDPM) [4] and One-Sided Stochastic Interpolants [2]. These algorithms can utilize the same neural model, which enables us to isolate the impact on quality of dynamics generation solely through the different generative transport algorithms. We find that our model significantly outpeforms other methods. Through these comparisons, we hope to demonstrate the benefits of our approach, and to clarify any concerns about the generative framework.
> - We would like to clarify that the reweighing of the configurations is only a different way of sampling batches from the training dataset, which should not affect the target interpolant learned by the model in the ideal case, but only the priority given to learning interpolants from different starting configurations. Since the model is only used to transition from one configuration to the next, it does not learn an "absolute" probability distribution for the configurations, but rather the conditional probability distribution $P(x_{t+1}|x_t)$. In this notation, while the reweighing biases the ($x_t$, $x_{t+1}$) pairs, it preserves the $x_{t+1}$ distribution for a given $x_t$. Having learned this, we could in principle reconstruct the Boltzmann distribution starting from a single configuration, by just repeatedly applying the evolution to reconstruct a very long MD trajectory. To optimize the sampling, we have decided instead to run many shorter trajectories. We agree with the reviewer that if we were to simply evolve from some biased starting points ($E(x_t)$), we would not obtain a Boltzmann distribution. However, as was done with the reference dataset, we use the sampled trajectories to build a Markov state model and reweigh our samples according to this. This procedure is independent of the starting distribution and only takes into account the transition probability: for this reason by correctly learning that, it allows us to reconstruct the Boltzmann distribution. We now clarify this in Section 4.1.
> - We made significant effort to obtain more statistical metrics and more comparisons across meaningful architectural and algorithmic axes. Please refer to Sections: (4.3) for comparison to additional generative frameworks across different noise levels; (4.4) for ablation on model capacity and quantitative metrics for the comparison against CG-MLFF; (4.41) for simulation performance estimates; and (Appendix 11) for a study on the effects of sampling parameters.
>
> We thank the reviewer again for their constructive feedback.
>
> > [1] Rezende et al. "Variational inference with normalizing flows." International conference on machine learning. PMLR, 2015.
> >
> > [2] Albergo et al. "Stochastic interpolants: A unifying framework for flows and diffusions." arXiv preprint arXiv:2303.08797 (2023
> >
> > [3] Lipman et al. "Flow matching for generative modeling." arXiv preprint arXiv:2210.02747 (2022).
> >
> > [4] Ho et al. "Denoising diffusion probabilistic models." Advances in neural information processing systems 33 (2020): 6840-6851.

---

> > ### Comment · Reviewer_zvV7 · 2024-11-29
> >
> > Thanks for the authors' response, which have addressed some of my concerns. I will take them into consideration.

---

### Official Review · Reviewer_sfuF · 2024-10-22

**Soundness:** 3
**Presentation:** 3
**Contribution:** 3
**Rating:** 6
**Confidence:** 5

**Summary:**

This paper proposes EquiJump for protein dynamics simulation. It utilizes two-sided stochastic interpolants to solve the issue of mismatch between  prior distribution and data distribution. The model is designed to meet SO(3) equivariance to capture the dynamics physically, and is evaluated on 12 fast-folding proteins, achieving state-of-the-art performance compared with CG-MLFF.

**Strengths:**

1. The proposal of  two-sided stochastic interpolants is inspiring and interesting. It enables the model to make sequence generation ($X^t\rightarrow X^{t+1}$), addressing the limitation of common (one-sided) stochastic interpolants, which are restricted to generating samples only from a specified prior distribution (e.g., a Normal Distribution).
2. The paper provides theoretical guarantee of EquiJump, which is verified by  the experimental results on fast-folding proteins. It successfully generates the  trajectories with longer timesteps. Moreover, the representation and visualization of the results are wonderful.

**Weaknesses:**

1. There are some typos:
  + In Eq. (6), the target to minimize should be $\hat{\eta}$，not $\hat{s}$
  + In line 94, "represents" and "builds" are repeated. Maybe you want to choose either of them?
  + In figure(3),  there are two subfigures (c). And in the final subfigure, the initial step is supposed to be $X^t$， not $X^\tau$.
2. The experiments can be more comprehensive:
  + More baselines. EquiJump is only compared with CG-MLFF. It will be more convincing to compare with predictive models, such as EGNN [1], EqMotion [2] and ESTAG[3]. They are recent equivariant graph neural networks (GNNs) and have been used  to predict molecular dynamics.
  + More representations. The experimental results are presented by plots and curves. Detailed numerical results (presented by table) can help us understand the priority of your model better.  For example, you can present the RMSD between your generated conformations and the ground truth structures.
  + More metrics. Jensen-Shannon divergence (JSD) is also used as a metric for trajectory prediction.
  + More datasets. I am curious about the performance of your model on the MDAnalysis [4] dataset, too.  It is also a prevailing protein dynamics dataset used for evaluating ML model.
  + Ablation on steps. Currently, you set the integration steps to 500. It is interesting to see the performance of fewer steps (e.g., steps=200/300/400).

[1] Satorras, Vıctor Garcia, Emiel Hoogeboom, and Max Welling. "E (n) equivariant graph neural networks." International conference on machine learning. PMLR, 2021.

[2] Xu, Chenxin, et al. "Eqmotion: Equivariant multi-agent motion prediction with invariant interaction reasoning." Proceedings of the IEEE/CVF Conference on Computer Vision and Pattern Recognition. 2023.

[3] Wu, Liming, et al. "Equivariant spatio-temporal attentive graph networks to simulate physical dynamics." Advances in Neural Information Processing Systems 36 (2024).

[4] Michaud‐Agrawal, Naveen, et al. "MDAnalysis: a toolkit for the analysis of molecular dynamics simulations." Journal of computational chemistry 32.10 (2011): 2319-2327.

**Questions:**

1. Can you provide more explanations about one-side and two-side stochastic interpolants? It is difficult to understand the contribution of your model if one has not much knowledge about it.
2. How many orders do you use for the feature representation $V$ ? And how do you initialize $V^0$ ($l=0$)?
3. Are there some trainable parameters of $I_\tau$?  From algorithm 1,  it seems that $I_\tau$ is just a linear interpolant, without anything to train.

---

> ### Author Response · Authors · 2024-11-21
>
> We thank the reviewer for their comments and questions.
>
> All the typos pointed out by the reviewer have been corrected.
>
> Here are some partial answers to the other points:
> - We thank the reviewer for pointing out these references, which we are adding to the main text.
> > - [1] Satorras, Vıctor Garcia, Emiel Hoogeboom, and Max Welling. "E (n) equivariant graph neural networks." International conference on machine learning. PMLR, 2021. This is a standard MLFF, so it can be used to do MD at 1fs timesteps. Other MLFF like MACE have been trained on different but possibly compatible all atom data. We included a section on the main text describing the performance comparison between these models.
> > - [2] Xu, Chenxin, et al. "Eqmotion: Equivariant multi-agent motion prediction with invariant interaction reasoning." Proceedings of the IEEE/CVF Conference on Computer Vision and Pattern Recognition. 2023.
> EqMotion is a general purpose motion predictor that can infer next steps given a number of previous steps. The only MD task presented is MD17, which are small molecules and fs timesteps. It is thus unclear whether it would be appropriate for the problem considered in this work since we are concerned with long time dynamics and not reproducing deterministically short time steps as EqMotion does.
> > - [3] Wu, Liming, et al. "Equivariant spatio-temporal attentive graph networks to simulate physical dynamics." Advances in Neural Information Processing Systems 36 (2024).
> ESTAG solves the problem of non-Markovian dynamics by considering multiple past frames. However, the focus of ESTAG is the deterministic reconstruction of trajectories, and it is thus hard to compare its results to the present approach. They show effectiveness on a protein trajectory, but no study is made on long-term behavior or stability. EquiJump instead focuses on learning stable long-time dynamics, and only has access to the current state.
>
> - We have added more quantitative measures of the quality of our trajectories in the form of new tables in the main text and appendix with several observables, including the RMSD as suggested by the reviewer. We also measure the difference between the free energy distributions in terms of JS divergence as recommended in order to quantify their quality:
>
> > The tables below compare  CG-MLFF and EquiJump, across several metrics. These are:
> > - **TICS**: Time-lagged Independent Component Analysis (TICA) is used to project molecular dynamics into a reduced dimensional space, emphasizing slow collective motions.
> > - **RMSD (Root Mean Square Deviation)**: Measures the average deviation of atomic positions from a reference structure.
> > - **GDT-TS (Global Distance Test - Total Score)**: A structural alignment metric evaluating the proportion of atoms in the model trajectory that are within a specified distance threshold of the reference structure.
> > - **RG (Radius of Gyration)**: Describes the compactness of the protein structure, measuring the spread of atoms around the protein's center of mass.
> > - **FNC (Fraction of Native Contacts)**: Fraction of interatomic contacts in the model trajectory that match those in the reference structure.
>
> >	Jensen-Shannon Divergence:
> > | Metric | CG-MLFF | EquiJump |
> > |---|---|---|
> > | TIC1 | 0.30 | **0.03** |
> > | TIC2 | 0.23 | **0.03** |
> > | RMSD | 0.20 | **0.03** |
> > | GDT-TS | 0.21 | **0.02** |
> > | RG | 0.18 | **0.04** |
> > | FNC | 0.27 | **0.03** |
>
> >Percent Error of Mean Observable Value:
> > | Metric | CG-MLFF | EquiJump |
> > |---|---|---|
> > | RMSD | 34.7 | **15.2** |
> > | GDT-TS | 57.1 | **18.3** |
> > | RG | 9.4 | **4.3** |
> > | FNC | 45.2 | **15.7** |
> >
>
> - While we are interested in the datasets of MDAnalysis, their trajectories have snapshots at the order of 1k samples, while the fast-folding dataset is at the order of 10M samples. Because our focus is in evaluating precise density distributions, we opted to investigate the dynamics of these proteins instead. However, we note that our model could readily be employed in these datasets, and we are thinking if there’s any quick experiment we could try.
> - We are working on adding an ablation on effects of variation of number of steps.
>
> Specific questions:
> 1. We are expanding the section on stochastic interpolants and are explaining the difference between the two settings (one vs two-sided interpolants). We are also working to obtain an experimental demonstration of their difference.
> 2. The representation V only goes up to order l=1. At initialization, the l=0 component of V contains an embedding of the (fixed) residue identity, while the l=1 contains information about the (changing) positions of the heavy atoms.
> 3. While I_\tau could be more general, it is in our case indeed just a linear interpolant.

---

> > ### Comment · Reviewer_sfuF · 2024-11-21
> >
> > Thank you for your response, which addresses some of my concerns. I look forward to your revised paper.

---

> ### Author Response · Authors · 2024-11-29
>
> We thank the reviewer for their thoughtful comments, and would like to update our response in light of the paper revision posted. In it, we made significant effort to address the following:
> - **More baselines**: we now provide extensive baselines across different generative frameworks and latent variable noise scales (Section 4.3), model size/capacity (Section 4.4), performance estimates (Section 4.4.1) and sampling parameters (Appendix 11). While we do not compare to other architecture directly, we now emphasize in Section (3.3.1) the use of established modules of Euclidean-equivariant networks and the focus of our work on the generative transport aspect of the problem (Section 4.3). To better situate the application of our model, we provide additional elaboration for the works discussed [1-3] on Related Work (Section 2).
> - **More representations**: we added a substantial number of comparative tables (Tables 1, 2, 3, 4, 6, Appendix 8) for detailed numerical results (e.g. RMSD of model samples from reference crystal). We also provide additional free energy curves across baselines and observables (Figure 4, Appendices 9, 10).
> - **More metrics**: we now measure diverse structural metrics (Appendix 7) and following the suggestion use Jensen-Shannon Divergence (JSD) for comparing them across baselines (Sections 4.3, 4.4, Appendix 8).
> - **Ablation on number of steps**: we added a new section (Appendix 11) for studying the effects of varying the number of sampling steps on sampling quality (Figure 15), additionally considering noising scales and providing qualitative plots for these effects on TIC space free energies.  Please note that in this updated set of experiments we use the default number of steps of 100.
> - **One-Sided Interpolant**: we increased our discussion of the One-Sided Interpolant in Section 3.1, and now directly compare it to our two-sided approach quantitatively and qualitatively in Section 4.3.
>
> We again thank the reviewer for the insightful feedback and hope to have adequately addressed the concerns raised.

---

> > ### Comment · Reviewer_sfuF · 2024-11-30
> >
> > In the revised manuscript, the authors added many experiments, including comparing with more baselines and using various metrics, which I think is enough to verify the effectiveness of EquiJump. Hence, I will raise my score.

---

### Official Review · Reviewer_hnmF · 2024-10-27

**Soundness:** 2
**Presentation:** 2
**Contribution:** 2
**Rating:** 3
**Confidence:** 4

**Summary:**

This paper proposes new methods for speeding up protein dynamics simulation using neural networks. The core novelty lies in the development of novel architectures termed EquiJump. This network is shown to recover protein dynamics as measured by trajectory error in 3D as well the evolution of free energy metrics over time.

**Strengths:**

A substantive assessment of the strengths of the paper, touching on each of the following dimensions: originality, quality, clarity, and significance. We encourage reviewers to be broad in their definitions of originality and significance. For example, originality may arise from a new definition or problem formulation, creative combinations of existing ideas, application to a new domain, or removing limitations from prior results. You can incorporate Markdown and Latex into your review. See https://openreview.net/faq.

The proposed architectural novelty is high with clear descriptions of the new elements. Although I suggest in figure 4, where the architecture is drawn to use consistent naming across the various panes, e.g. what is called the ‘deep network’ in panel (c) becomes the ‘conditioner’ in panel (d) …. Also two panel c’s appear in the figure etc.

The experiments are done on a dataset of 12 folding proteins, which appear to a standard point of comparison.

I appreciated the representation of molecular data using geometric features, although it was unclear what the irreducible representation is here (Lie algebraic coordinates?).

**Weaknesses:**

Although the architecture is novel, it is less clear how it is motivated, and why this particular combination works. This also ties in to comparisons, as only one primary comparison point is used – CGMLFF.
Rotation data representation was unclear – ‘irrep’ would benefit by a clearer definition. Does using special approaches like this, does it reduce the parameterization of the rest of the network?


It would be also important to consider model size when comparing to other models, e.g.  total number of parameters of the network for CGMLFF is noted as 294,565 in (Majewski 2023). Would also be useful to see training curves for the models.

Training-validation-test splits are not clearly specified from what I can tell. (Majewski 2023) states the data was randomly split between training (85%), validation (5%), and testing (10%).

Improvements in performance are mostly left as visual assessment, e.g. figure 6 and figure 5. It is unclear how one would conclude anything just by looking at these figures. Multiple quantitative ways to compare seem feasible, e.g. see supplementary docs in (Majewski 2023).

https://static-content.springer.com/esm/art%3A10.1038%2Fs41467-023-41343-1/MediaObjects/41467_2023_41343_MOESM1_ESM.pdf

**Questions:**

Clarify the training-test-validation splits.
Comment on performance comparison approach more quantitatively (not needing new experiments).
Model size comparison.
Comment on what the specific rotation representation is, and how does it actually help. Also, some insight on how the network archietcture for EquiJump can be motivated.

---

> ### Author Response · Authors · 2024-11-21
>
> We thank the reviewer for their questions and remarks.
>
> We are working to strengthen all the aspects mentioned by the reviewer. Here are some preliminary comments:
>
> 1. We clarified the use of irreps in the paper. These are irreducible representations of SO(3), and can be understood as the network having feature representations in bases of spherical harmonics. This representation of geometric features and 3D coordinates is motivated for enabling compact all-atom representation of proteins at the level of residues. This contrasts with having a full feature representation per atom, as in most atomic GNN-style models. Similarly, the architecture is primarily motivated from intersecting this type of 3D representation with two-sided stochastic interpolants. The four headers arise from (2 types of 3D data) x (2 predictions of noise and drift of the interpolants). We are adding updates to their description to clarify these motivations.
> 2. We are adding a section to compare the size and performance of the model sizes we investigated (6M-390M) to CGMLFF (294k): while our network has a much larger number of parameters, it performs even larger (10^5 times larger) step sizes compared to the MLFF at overall shorter wall time per step as discussed in the time analysis. We argue that the different approach to the simulation justifies the need for a larger model, resulting nevertheless in a faster inference time:
> > - Direct performance comparisons between methods are challenging, but we can provide estimates. For the largest protein in our study, lambda (12,000 atoms with explicit water), Amber22 benchmarks on NVIDIA A100 hardware indicate that a 100 ps timestep would take ~3.6 seconds. In contrast, EquiJump samples the phase space in parallel, completing a 100 ps step for a single simulation in just 0.47 seconds—a 7.5x speed-up. This efficiency could improve further with optimized hyperparameters.
> > - CG-MLFF, while lacking explicit performance figures, is estimated to be 1–2 orders of magnitude slower than classical MD due to its computationally intensive potential. Similarly, state-of-the-art MLFFs like MACE-OFF (trained on biological data) require ~860 seconds for a 100 ps timestep on a 1200-atom system, making them impractical for this task. These comparisons emphasize EquiJump's exceptional efficiency and its suitability for rapid phase space exploration in large-scale systems.
> 3. The splitting of the data in train/validation/test splits is a common practice to evaluate deterministic networks, where a correct output for each input can be identified. However, in the context of generative models, where the output of the network is sampled from a distribution, it is much harder to evaluate its quality, for lack of a single correct answer. Since our model aims at reproducing the long term dynamic of a system in contact with a thermostat, it falls under this category, and it is impractical to assess its generalization via conventional splits. Instead, our approach relies on statistical properties, such as free energy distributions and trajectory comparisons, to validate model performance.
> 4. We have added more quantitative analysis of our results in the form of new tables in the main text and appendix with observables similar to the ones pointed out by the reviewer. We also measure the difference between the free energy distributions in terms of JS divergence in order to quantify their quality:
> > The tables below compare CG-MLFF and EquiJump, across several metrics. These are:
> > - **TIC1 & TIC2**: Time-lagged Independent Component Analysis (TICA) is used to project molecular dynamics into a reduced dimensional space, emphasizing slow motions.
> > - **RMSD (Root Mean Square Deviation)**: Measures the average deviation of atomic positions from a reference structure.
> > - **GDT-TS (Global Distance Test - Total Score)**: A structural alignment metric evaluating the proportion of atoms in the model trajectory that are within a distance threshold of the reference structure.
> > - **RG (Radius of Gyration)**: Describes the compactness of the protein structure, measuring the spread of atoms around the protein's center of mass.
> > - **RMSF (Root Mean Square Fluctuations)**: Measures atomic positional fluctuations over time, capturing the flexibility of regions in the protein.
> > - **FNC (Fraction of Native Contacts)**: Fraction of interatomic contacts in the model trajectory that match those in the reference structure.
>
>
> >	Jensen-Shannon Divergence:
> > | Metric | CG-MLFF | EquiJump |
> > |---|---|---|
> > | TIC1 | 0.30 | **0.03** |
> > | TIC2 | 0.23 | **0.03** |
> > | RMSD | 0.20 | **0.03** |
> > | GDT-TS | 0.21 | **0.02** |
> > | RG | 0.18 | **0.04** |
> > | FNC | 0.27 | **0.03** |
>
> >Percent Error of Mean Observable Value:
> > | Metric | CG-MLFF | EquiJump |
> > |---|---|---|
> > | RMSD | 34.7 | **15.2** |
> > | GDT-TS | 57.1 | **18.3** |
> > | RG | 9.4 | **4.3** |
> > | FNC | 45.2 | **15.7** |

---

> > ### Comment · Reviewer_hnmF · 2024-11-24
> >
> > Thank you the responses; I will take these into consideration.

---

> ### Author Response · Authors · 2024-11-29
>
> We thank the reviewer for their thoughtful comments and would like to update our response in light of the revised paper:
> - **Model Size Comparison**: We added experiments (Section 4.4, Appendix 9) for better understanding the behavior of our model and its quality in relation to model size, using meaningful structural metrics (see below). Based on that, we added a section for discussing our model across capacities and its relation to CG-MLFF and force-field models (Section 4.4). While we found our model to perform competitively in low capacity regimes similar to CG-MLFF, our approach really excelled in high capacity sizes. We explain that by noting that our model steps at considerably higher time rates, while noting its increased comparative performance. We additionally estimate performance ablations (Section 4.4.1) discussing the acceleration benefits of the larger stepping and the impact of model size on speedup.
> - **Quantitative Metrics**: we followed the suggested supplementary text of (Majewski 2023) and made significant effort to add strong quantitative results in the paper (Section 4.4; Appendix 8; Tables 2, 3, 4, 6), giving solid numbers and complementing the visual assessment of Figures 6 and (originally, 5, now 12 and additionally 13, 14). Using these quantities and different model sizes, we estimated the trade-off between acceleration and generation quality in Section 4.4.1. Finally, following the same metrics (Table 1) we added a new section (Section 4.3) with performance comparisons between generative transport models where we demonstrate the benefits of our method. We hope this additional thorough comparison addresses the reviewer’s concerns and further highlights the novelty and practical significance of our approach.

---

### Official Review · Reviewer_Jvpj · 2024-11-02

**Soundness:** 3
**Presentation:** 3
**Contribution:** 3
**Rating:** 6
**Confidence:** 4

**Summary:**

In the paper, the authors introduce the EquiJump framework to generate protein dynamics simulation trajectories. Based on the stochastic interpolants framework, the authors capture the conditional probability distribution at different time to generate the whole trajectory. Additionally, the paper extend the interpolants on geometric features and design new network architecture. Experiments on 12 fast-folding proteins show the empirical performance of the framework.

**Strengths:**

1. The presentation of the method is very clear and easy to understand.

2. To the best of my knowledge, the modification of the stochastic interpolants framework for generating protein dynamics is novel.

3. Employing stochastic interpolants to generate trajectories is a well-founded and rational approach.

**Weaknesses:**

1. I believe it is essential to conduct comparative analyses with other baseline methodologies to establish the efficacy of your approach. For instance, utilizing a Machine-Learned Force Field (MLFF) that incorporates your network architecture could serve as a viable baseline. Such a comparison could demonstrate that your interpolants method outperforms the MLFF, independent of the network design, thereby highlighting the strengths of your technique.

2. I have observed that the sampling process within the current framework is rather inefficient and time-consuming. Therefore, it would be beneficial to include a comprehensive comparison of computational costs between the EquiJump framework and preceding MLFF methods. Such an analysis would be instrumental in assessing the practicality and efficiency of the EquiJump framework in relation to established techniques.

3. It is also crucial to engage in a detailed discussion on contemporary methods of trajectory generation. References [1-4] represent concurrent studies in trajectory generation. Providing a clear delineation of the distinctions and relationships between your work and these existing studies will help elucidate the unique contributions and advancements your research offers to the field.

[1]. Du, Yuanqi, et al. "Doob's Lagrangian: A Sample-Efficient Variational Approach to Transition Path Sampling." arXiv:2410.07974 (2024).

[2]. Zhang, Xi, et al. "Trajectory Flow Matching with Applications to Clinical Time Series Modeling." arXiv:2410.21154 (2024).

[3]. Luo, Shengjie et al. “Bridging Geometric States via Geometric Diffusion Bridge.” arXiv:2410.24220 (2024).

[4]. Han, Jiaqi, et al. "Geometric Trajectory Diffusion Models." arXiv:2410.13027 (2024).

**Questions:**

See the weaknesses section above.

---

> ### Author Response · Authors · 2024-11-21
>
> We thank the reviewer for their comments.
>
> While we're working to improve on the points raised by the reviewer, here are some preliminary answers:
> 1. We have added a theoretical performance comparison to a realistic MLFF (see point 2 below), and show our method surpasses it in both performance and efficiency.  However, obtaining an experimental baseline with an MLFF on this dataset is particularly challenging, as it requires all-atom force information at each timestep, which is unavailable. MLFFs rely heavily on force-matching during training, and without this data, direct training of an MLFF is not feasible. As seen in the case of  CGMLFF (in a coarse-grained fashion) any simulation with MLFF would require timesteps close to the original dynamics (few fs) and precise force matching, making direct training of a MLFF on these data unfeasible. Moreover, as the reviewer points out, we believe that the main achievement of our work lies in the stochastic interpolants method and the possibility to bypass the whole dynamics. This is exemplified by the fact that we surpass both in efficiency (point 2 below) and accuracy CGMLFF, which are inherently faster than MLFF (without coarse graining). While we show that the proposed architecture is fit for this task, its main structure is not very different from other networks used in MLFF tasks (e.g. MACE, NequIP, etc.). We are also working on a comparison between this stochastic interpolant formulation and other generative methods (e.g. flow matching) for comparison.
> 2. We have added a new section to the main paper highlighting the inference cost of our method compared to the original simulation, CGMLFF, and a commonly used MLFF:
> > In terms of performance, direct comparisons between different methods are not entirely straightforward, but we can provide estimates to contextualize their relative speeds. Taking the largest protein in our study, lambda, as a reference: the original simulations with explicit water involve approximately 12,000 atoms, as described in (Lindorff Larsen et al. 11’). According to Amber22 benchmarks on the same hardware (NVIDIA A100), a system twice this size (JAC) achieves a throughput of 1258 ns/day. Scaling this performance linearly to match the size of the lambda system, a single 100 ps timestep would take approximately 3.6 seconds. In contrast, with EquiJump, we can sample the phase space in parallel, running a batch of 100 simulations on a single card in 47 seconds. This translates to just 0.47 seconds per 100 ps step for a single simulation, providing a 7.5x speed-up compared to the original simulation. Furthermore, this implementation has not been optimized for speed, and tuning hyperparameters—such as the target evaluation time and the number of diffusion steps—could enhance performance even further. For CG-MLFF, while explicit performance figures are not reported, it is estimated to be 1–2 orders of magnitude slower than classical molecular dynamics simulations with explicit water. This is expected, as CG-MLFF operates with timesteps comparable to classical MD but uses a more computationally intensive potential, even at the coarse-grained level. Finally, for a modern MLFF like MACE-OFF (Kovacs et al. 23’), which represents state-of-the-art performance and has been trained on biological data, we can estimate the time required for a similar simulation. For a 1200-atom system on the same hardware, MACE-OFF achieves approximately 2.5M steps per day for the smallest model. With a 4 fs timestep (comparable to classical MD), this equates to roughly 860 seconds for a single 100 ps timestep, making such simulations infeasible for the present task. These comparisons illustrate the significant efficiency of EquiJump, particularly for large-scale systems, and highlight its suitability for tasks requiring rapid phase space exploration.

---

> ### Author Response · Authors · 2024-11-21
>
> 3. We thank the reviewer for pointing out these works, we are now including them in section 2 of the paper.
> > EquiJump distinguishes itself in the context of these related works by its unique capability to model long-time dynamics probabilistically across the entire phase space. Here's a detailed comparison:
> > - [1] Du et al. (2024): "Doob's Lagrangian"
> This work provides an efficient method for sampling transition paths between two states (A and B) using a neural network trained on a custom objective. While highly efficient for specific path sampling, it is inherently limited to targeted transitions and does not aim to capture the Boltzmann distribution over the entire phase space. In contrast, EquiJump is designed to represent the global dynamics of the system, allowing sampling across the phase space rather than focusing solely on specific paths. Additionally, EquiJump achieves this with significantly fewer evaluations compared to the 25M evaluations reported by Du et al. for a single 1 ps path.
> > - [2] Zhang et al. (2024): "Trajectory Flow Matching"
>  This paper uses flow matching to predict the next state in a time series based on previous steps, but it focuses on deterministic predictions with relatively small timesteps. Physical examples in the paper are limited to simple systems like a harmonic oscillator, which lack the complexity of molecular systems. EquiJump, on the other hand, excels in probabilistic predictions over long timesteps and is applied to realistic molecular systems like proteins, capturing the stochastic nature of such dynamics without requiring fine-grained temporal resolutions.
> > - [3] Luo et al. (2024): "Geometric Diffusion Bridge (GDB)"
> GDBs aim to evolve systems between two states and are presented as a general method. However, their use case is primarily shown for equilibrium or relaxed states, and they do not demonstrate evolution through a trajectory or long-time dynamics. In contrast, EquiJump is explicitly designed to generate meaningful trajectories over extended timescales, bridging states while maintaining compatibility with the full stochastic nature of molecular dynamics.
> > - [4] Han et al. (2024): "Geometric Trajectory Diffusion Models (GeoTDM)" GeoTDM provides a general framework for generating constrained or unconstrained trajectories, including MD tasks. However, it operates on timescales typical of MLFFs, requiring small timesteps and often producing deterministic trajectories. EquiJump surpasses this limitation by enabling long-timestep probabilistic predictions, allowing efficient exploration of the phase space without the computational burden associated with fine-grained simulations.

---

> > ### Comment · Reviewer_Jvpj · 2024-11-25
> >
> > Thank you for your comprehensive response. While your reply has addressed some of my concerns, I find myself in disagreement with your analysis of the related work cited [1-4]. In my view, the methodologies employed in references [2] and [3] are very similar to your own. The principal distinction lies in their use of a flow matching vector field as opposed to the stochastic interpolant vector field that you have implemented. It is my belief that each of these methods is theoretically capable of generating stochastic dynamics, even though the experimental data they may have been trained on could be deterministic in nature.
> >
> > For instance, the work in [2] leverages a flow matching framework to predict subsequent states, which echoes the methodology in your study—this is especially evident in Section 3 of their paper. They concentrate on time-series data analysis. Similarly, [3] is adept at tracing the progression or trajectory of geometric states, as detailed in Section 3.2 of their publication, and its focus is on the evolution of molecular systems towards equilibrium or relaxed states. [4] also examines relatively small molecular systems. Consequently, although the underlying methodologies are comparable, the experimental frameworks and datasets you have utilized are distinct. This constitutes a second, and arguably more significant, difference.
> >
> > A meticulous comparison that highlights the differences and similarities between your work and the existing literature would not only clarify your distinctive contributions but would also enhance the overall understanding of your study's positioning within the field.

---

> ### Author Response · Authors · 2024-11-29
>
> We thank the reviewer for their thoughtful comments and would like to clarify the positioning of our work relative to [2] (Trajectory Flow Matching), [3] (Geometric Diffusion Bridge), and [4] (Geometric Trajectory Diffusion Models):
>
> In our understanding, [2] employs Flow Matching (FM) [5] for time evolution, while using noise terms for uncertainty quantification in irregular time-series data. Our approach implements Stochastic Interpolants [6] which generalize Flow Matching by having both a drift and a noise term for the transport. Moreover, while [2] is applied to clinical time-series modeling, EquiJump is implemented for the dynamics of protein systems.
>
> Both [3] and [4] are similar to our model, as they focus on direct transport from the previous step to the next. While [3] and [4] implement this transport through a denoising diffusion-based [7] approach, learning the score for moving samples, our model uses Two-Sided Stochastic Interpolants [6] and uses both drift and score (in our framing, noise) terms that further generalize diffusion. [3] additionally studies trajectory guidance and multi-step systems, being ultimately applied to predicting equilibrium states and performing structural relaxation of small molecules. [4] focuses on general 3D tasks such as N-body systems, MD of small molecules and pedestrian trajectory forecasting. In contrast, our work is focused on the task of long-term dynamics on 12 large proteins. To summarize, our work proposes step-to-step direct generative transport of samples for dynamics simulation in a similar fashion to concurrent [3] and [4], while implementing a more general framing based on Stochastic Interpolants [6] that also implements drift. Our works also differ in representation (protein vs. small molecules, residue-based vs atom-based) and in application, where we are interested in reproducing precise time dynamics on a large-scale MD dataset.
>
> We added further discussion about these considered papers [1-4] in the Related Work section, highlighting how they situate in relation to our work.
>
> Finally, to better compare our model to other generative transport methods, we trained different generative frameworks and compared it to the Two-Sided Interpolant of EquiJump in Section 4.3.
>
> In this review, we additionally included a thorough analysis of EquiJump’s performance in terms of model size (Section 4.4), performance (Section 4.4.1) and sampling hyperparameters (Appendix 11).
>
> We thank the reviewer again for the insightful discussion and hope to have effectively addressed the valuable feedback provided.
>
> > [1]. Du, Yuanqi, et al. "Doob's Lagrangian: A Sample-Efficient Variational Approach to Transition Path Sampling." arXiv:2410.07974 (2024).
> >
> > [2]. Zhang, Xi, et al. "Trajectory Flow Matching with Applications to Clinical Time Series Modeling." arXiv:2410.21154 (2024).
> >
> > [3]. Luo, Shengjie et al. “Bridging Geometric States via Geometric Diffusion Bridge.” arXiv:2410.24220 (2024).
> >
> > [4]. Han, Jiaqi, et al. "Geometric Trajectory Diffusion Models." arXiv:2410.13027 (2024).
> >
> > [5] Lipman, Yaron, et al. "Flow matching for generative modeling." arXiv preprint arXiv:2210.02747 (2022).
> >
> > [6] Albergo, Michael S., Nicholas M. Boffi, and Eric Vanden-Eijnden. "Stochastic interpolants: A unifying framework for flows and diffusions." arXiv preprint arXiv:2303.08797 (2023).
> >
> > [7] Ho, Jonathan, Ajay Jain, and Pieter Abbeel. "Denoising diffusion probabilistic models." Advances in neural information processing systems 33 (2020): 6840-6851.

---

> > ### Comment · Reviewer_Jvpj · 2024-11-29
> >
> > Thank you for your comprehensive response, which addresses all of my concerns. I increase my score to 6.

---

### Official Review · Reviewer_hezM · 2024-11-03

**Soundness:** 3
**Presentation:** 3
**Contribution:** 3
**Rating:** 6
**Confidence:** 2

**Summary:**

The authors present a novel equivariant network that predicts  states from an MD trajectory. Presented results look convincing, but unfortunately the State of Reproducibility is not written and the model/code are not available for evaluation.

**Strengths:**

The network generalizes on multiple protein families and outperforms the SOTA results.

**Weaknesses:**

It will be useful to assess the stereochemical quality of the produced intermediate protein structures using standard metrics, ProCheck and/or Molprobity.

A minor point -- It will be also useful to formally prove SO(3) equivariance of the architecture blocks (without reading the original papers).

**Questions:**

Is it possible to check if the simulated trajectory stays on a closed manifold in the phase space and evaluate its (shadow) Hamiltonian? Would it make sense?  Please try to estimate the phase space volumes and/or energy fluctuations over time.

Please provide ProCheck/Molprobity Ramachandran plot statistics, bond lengths/angles, clash scores, and overall statistics. Please compare these to the values computed on the MD trajectories.

Please provide mathematical proofs or empirical demonstrations of equivariance for the key components of their architecture that are not directly taken from prior work, otherwise please explain the key equivariance components of the prior work, such as equivariant linear layers, etc.

---

> ### Author Response · Authors · 2024-11-21
>
> We thank the reviewer for their questions and comments.
>
> We are actively working on all the points raised by the reviewer, here are some preliminary answers:
>
> - The TICA plots presented in the paper are a low dimensional confirmation that our simulations stay in the manifold of the original MD simulations. To obtain a more precise measure of this, we estimated several observables over our trajectories and showed that the distribution matches the one from the original data, please refer to the new tables in the main text and appendix material. Given the fact that we are dealing with a system in contact with a thermostat, and only producing configurations at a very long time step of 100ps, we do not believe that it is possible to evaluate a (shadow) Hamiltonian for this system. We are open to do this if the reviewer could suggest a relevant reference.
> - We are working on obtaining ProCheck/Molprobity and similar metrics that can further demonstrate the chemical validity and manifold stability of our method.
> - We are elaborating further on Euclidean equivariance and adding a section to the Appendix of the paper mathematically proving that all the operations in our network preserve it.

---

> ### Author Response · Authors · 2024-11-29
>
> We would like to thank the reviewer again and provide some highlights about the latest version of the manuscripts related to their comments:
> - We added a new Appendix (A.6) studying the chemical validity of EquiJump samples. Following the suggestions, we added measures for sampled bond lengths, bond angles and estimated number of clashes based on van der Waals radii.
> - Additionally, Tables 2 and 3, and appendix A.8 now report several quantitative measures of the quality of our trajectories, addressing the statistical similarity to the ground truth across diverse structural metrics. Finally, Fig.5 and 9 report more qualitative measures, and Figures 6 and 10 show that our models stay within the valid configuration space, in contrast with some of the failure cases of the ablation studies (Fig.4, 10 and 15).
> - We added a new section (Appendix section 2.1) on the appendix discussing and demonstrating the equivariance of our network.

---

### Author Response · Authors · 2024-11-29
**Summary of Revision Changes**

We express our sincere gratitude to all reviewers for their valuable feedback. We have diligently addressed concerns and implemented the suggested improvements, significantly enhancing the manuscript. We summarize key changes in the revised manuscript below:

- **Comparison of Generative Transport Methods**: in new Section 4.3 we compare our model’s generative framework to other generative transport methods (Figure 2) [1, 2, 3]. We quantitatively evaluate the performance of these transport models across noise scales (Table 1) and qualitatively compare their free energies (Figure 4). Our results demonstrate the efficacy of our proposed approach to MD snapshot data.
- **Model Size Ablation and Quantitative Comparisons**: in updated Section 4.4 and Appendices 8 and 9 we now ablate our model performance across model capacities and against CG-MLFF [4] with extensive quantitative measures. These experiments demonstrate the notable precision of our model in large-capacity regimes.
- **Analysis of Chemistry**: in Appendix 6 we evaluate the performance of EquiJump in generating chemically valid data across different proteins. We show the distribution of measures for chemical validity (bond length, bond angles, atomic clashes) for model samples and reference data, concluding that our model performs chemically stable simulation.
- **Performance Estimates**: in Section 4.4.1 we estimate simulation speedups relative to the original MD method that created the training data, while discussing performance of comparable models. We study the behavior of our approach across capacities, and find that EquiJump enables simulation acceleration with small loss of long-term dynamics accuracy.
- **Sampling Parameters Ablation**: in Appendix 11 we now investigate the impact in sampling of changing transport hyperparameters (number of steps, noise schedule). Our results are promising for further reduction of the number of sampling steps.
- **Structural Metrics**: following revision suggestions, we added diverse metrics (RMSD, GDT-TS, Radius of Gyration, Fraction of Native Contacts) for quantifying the performance of our model in relevant structural properties, and now provide strong statistical comparisons (Jensen-Shannon Divergence) for different model benchmarks. The summarized results described above make extensive use of those scores through tables 1, 2, 3, 4, and 6.
- **Expanded Related Work**: we further developed Section 2 with updated elaboration about the discussed similar work [5-11], highlighting their contributions and primary differences to our approach.
- **Expanded Stochastic Interpolants**: we added further description on One-Sided Interpolants in Section 3.1, and added it to the generative comparison of Section 4.3.
- **Additional Free Energy Curves**:  we provide additional estimated free energy curves for Radius of Gyration and Fraction of Native Contact in Appendix 10, Figures 13 and 14.

We thank the reviews again for the constructive discussion, and hope the new updates and results can demonstrate the utility of our work to the field.

> [1] Lipman, Yaron, et al. "Flow matching for generative modeling." arXiv preprint arXiv:2210.02747 (2022).
>
> [2] Ho, Jonathan, Ajay Jain, and Pieter Abbeel. "Denoising diffusion probabilistic models." Advances in neural information processing systems 33 (2020): 6840-6851
>
> [3] Albergo, Michael S., Nicholas M. Boffi, and Eric Vanden-Eijnden. "Stochastic interpolants: A unifying framework for flows and diffusions." arXiv preprint arXiv:2303.08797 (2023).
>
> [4] Majewski, Maciej, et al. "Machine learning coarse-grained potentials of protein thermodynamics." Nature communications 14.1 (2023): 5739.
>
> [5]. Du, Yuanqi, et al. "Doob's Lagrangian: A Sample-Efficient Variational Approach to Transition Path Sampling." arXiv:2410.07974 (2024).
>
> [6]. Zhang, Xi, et al. "Trajectory Flow Matching with Applications to Clinical Time Series Modeling." arXiv:2410.21154 (2024).
>
> [7]. Luo, Shengjie et al. “Bridging Geometric States via Geometric Diffusion Bridge.” arXiv:2410.24220 (2024).
>
> [8]. Han, Jiaqi, et al. "Geometric Trajectory Diffusion Models." arXiv:2410.13027 (2024).
>
> [9] Satorras, Vıctor Garcia, Emiel Hoogeboom, and Max Welling. "E (n) equivariant graph neural networks." International conference on machine learning. PMLR, 2021.
>
> [10] Xu, Chenxin, et al. "Eqmotion: Equivariant multi-agent motion prediction with invariant interaction reasoning." Proceedings of the IEEE/CVF Conference on Computer Vision and Pattern Recognition. 2023.
>
> [11] Wu, Liming, et al. "Equivariant spatio-temporal attentive graph networks to simulate physical dynamics." Advances in Neural Information Processing Systems 36 (2024).

---

### Meta-Review · Area_Chair_T7dD · 2024-12-22

**Metareview:**

The paper is on using machine learning, particularly generative modeling, for molecular dynamics. It uses Albergo et al’s (ICML 2024) stochastic interpolants on SO3 equivariant models to simulate 3D conformation state transfer of an MD for 12 proteins with relatively long time steps.

The reviewers found the presentation generally good despite some typos and unclarities, the proposed method reasonable for the stated purpose, the results significant compared to one included MLFF baseline.

On the other hand, they were concerned about the quality of the produced trajectory, lack of enough quantitative scalar measures, lack of a comprehensive conceptual and empirical comparison with other recent works on modeling MD trajectories including recent MLFF works or other generative modeling approaches, and they requested more diverse set of benchmark and evaluations including for out-of-distribution proteins, as well as additional side and ablation studies including a separation of backbone model/representation and the specific generative model.

The authors provided a thorough rebuttal and major update of the paper which improved several aspects including more empirical evaluation of the chemical validity of the generated individual states and full trajectory, a formal result of equivariance of the used model,  a more detailed and comprehensive conceptual comparison with other recent works, tables with quantitative single scalar metrics for the quality of the generated trajectory, and some additional basic baselines such as DDPM.

The AC believes the paper had clear merits in its initial form and has further considerably improved during the rebuttal. However, the AC recommends rejection for two main reasons: (1) the paper’s empirical evidence needs to be strengthened: since the main contribution of the paper is the use of a more proper modeling with stochastic interpolants for a specific application of long-term prediction of MD trajectories, it needs to (a) find a way to compare with other well-designed generative models to show the importance of the arbitrary prior and (b) evaluate on more benchmarks including OoD data to enable a solid progress in the new line they open for ML modeling of long steps with large molecules. (2) the paper requires another round of review: the paper has been majorly updated during the rebuttal and needs some more additions, therefore it requires a fresh round of review of the details of all new textual and formal discussions, evaluations, and empirical results.

**Additional Comments On Reviewer Discussion:**

The paper was reviewed by a panel of 5 experts covering the application, bioinformatics, molecular dynamics, generative modeling, geometric representation, and generally machine learning. They were all engaged in the discussion with the authors and among themselves. All reviewers rated the paper as borderline which necessitated more discussions after the rebuttal and a careful consideration of all materials by the AC before making the decision.

---

### Decision · Program_Chairs · 2025-01-22

Reject